# Septin7 is indispensable for proper skeletal muscle architecture and function

**Mónika Gönczi[1], Zsolt Ráduly[1,2], László Szabó[1,2], János Fodor[1], Andrea Telek[1], Nóra Dobrosi[1], Norbert Balogh[1,2], Péter Szentesi[1], Gréta Kis[3], Miklós Antal[3], György Trencsenyi[4], Beatrix Dienes[1], László Csernoch[1]***

[1]Department of Physiology, Faculty of Medicine, University of Debrecen, Debrecen, Hungary; [2]Doctoral School of Molecular Medicine, University of Debrecen, Debrecen, Hungary; [3]Department of Anatomy, Histology and Embryology, Faculty of Medicine, University of Debrecen, Debrecen, Hungary; [4]Division of Nuclear Medicine and Translational Imaging, Department of Medical Imaging, Faculty of Medicine, University of Debrecen, Debrecen, Hungary

**Abstract** Today septins are considered as the fourth component of the cytoskeleton, with the Septin7 isoform playing a critical role in the formation of higher-order structures. While its importance has already been confirmed in several intracellular processes of different organs, very little is known about its role in skeletal muscle. Here, using Septin7 conditional knockdown (KD) mouse model, the C2C12 cell line, and enzymatically isolated adult muscle fibers, the organization and localization of septin filaments are revealed, and an ontogenesis-dependent expression of Septin7 is demonstrated. KD mice displayed a characteristic hunchback phenotype with skeletal deformities, reduction in in vivo and in vitro force generation, and disorganized mitochondrial networks. Furthermore, knockout of Septin7 in C2C12 cells resulted in complete loss of cell division while KD cells provided evidence that Septin7 is essential for proper myotube differentiation. These and the transient increase in Septin7 expression following muscle injury suggest that it may be involved in muscle regeneration and development.

***For correspondence:**
csl@edu.unideb.hu

**Competing interest:** The authors declare that no competing interests exist.

## Editor's evaluation

This work combines a novel mouse model of inducible skeletal muscle specific deletion of Septin7 with Septin7 manipulation in vitro to explore the role of Septin7 in striated muscle development and function. The work should be of broad interest to muscle and cytoskeletal biologists as it indicates an essential role of Septin7 in normal muscle development and suggests potential roles in muscle regeneration as well.

## Introduction

Proper contractile activation of the striated muscle requires precisely orchestrated machinery consisting of the T-tubule with the embedded dihydropyridine receptor, the terminal cisternae of the sarcoplasmic reticulum (SR) with the ryanodine receptor (RyR) in its membrane, and mitochondria. Furthermore, appropriate localization of ion channels and proteins contributing to calcium homeostasis and downstream signaling pathways requires a precisely organized structure and coordinated function of the cytoskeletal networks providing the connection between intracellular organelles (e.g., mitochondria) within the sarcomere, and the linkage between sarcomeres and surface membrane. Recent studies have suggested a role of septins (*Gönczi et al., 2021*) in these key cellular processes in

cardiac and skeletal muscle function as well. However, muscle-specific function of the different septin isoforms has not yet been completely explored.

Septins have been first described as regulators of cytokinesis and cell polarity in yeast, and since their discovery (*Hartwell, 1971*) their expression has been demonstrated in many other organisms, including mice and humans. Numerous studies imply the important role of septins in several intracellular processes as molecular scaffolds and diffusion barriers that control localization of membrane proteins. They are also involved in host–pathogen interactions during infections, in cell mobility (*Kelley et al., 2015*), apoptosis (*Zhang et al., 2016*; *Fung et al., 2014*), endocytosis (*Hall and Russell, 2012*; *Barve et al., 2018*), determining cell shape (*Mostowy et al., 2011*), and even in mechanotransductional pathways (*Caudron and Barral, 2009*; *Gladfelter et al., 2001*; *Lam and Calvo, 2019*; *Mostowy and Cossart, 2012*). Since septins interact with actin, microtubules, and membrane structures of cells and assemble into filaments, they became generally accepted as the fourth cytoskeletal component (*Mostowy and Cossart, 2012*). However, the molecular mechanism of these interactions is still the focus of intense investigation (*Spiliotis and McMurray, 2020*; *Spiliotis and Nakos, 2021*; *Woods and Gladfelter, 2021*).

It is already known that these 30–65 kDa highly conserved GTP-binding proteins form heterooligomeric complexes, and besides filaments, they polymerize into higher-order structures, sheets, rings, and cage-like formations contributing to biological processes (*Weirich et al., 2008*; *Bertin et al., 2010*). The number of septin isoforms encoded is extremely variable among different organisms. In humans, 13 different isoforms were identified, which were classified into four groups (SEPTIN 2, 3, 6, and 7) on the basis of sequence homology (*Hall et al., 2005*), with a lone presence of Septin7 in its group. Their ubiquitous (*Hall et al., 2005*; *Cao et al., 2007*; *Connolly et al., 2011*) and/or tissue-specific expression influences the localization and function of cell surface proteins (*Caudron and Barral, 2009*; *Spiliotis and Nelson, 2006*). The altered expression of septins has been linked to neurodegenerative or excretory system diseases, cardiovascular failures, immunological problems, or cancer (*Dolat et al., 2014*).

Septin filaments are predominantly made up of hexamers and octamers of septins belonging to three or all four of homology groups, according to the cell type. All of these filaments include the Septin7, which indicates the pivotal role of this ubiquitous protein in the formation of heterooligomeric complexes described to date (*Sirajuddin et al., 2007*; *Jiao et al., 2020*; *DeRose et al., 2020*; *Soroor et al., 2021*; *Mendonça et al., 2021*). The knockout of Septin7 is embryonic lethal. Its absence resulted in the loss of other septin proteins in the oligomers (*Kinoshita et al., 2002*). Recently, the role of Septin7 has been demonstrated in nervous and reproductive systems and its diverse functions in various neurological diseases (Alzheimer's disease, schizophrenia, neuropsychiatric systemic lupus erythematosus), in the development of cancer (glioma, papillary thyroid carcinoma, and hepatocellular carcinoma). For a comprehensive review, see *Wang et al., 2018*. Furthermore, Septin7 has been proposed as a novel regulator of neuronal $Ca^{2+}$ homoeostasis (*Deb and Hasan, 2016*; *Deb et al., 2016*). It has been revealed to downregulate the expression of the Orai and inositol-trisphosphate-receptor 3 (IP3R), subsequently affecting the cytosolic $Ca^{2+}$ by which it can cause deficient flight ability in *Drosophila melanogaster* (*Deb et al., 2016*; *Tada et al., 2007*). Knockdown of Septin7 resulted in myofibrillar disorganization and functionally reduced ventricular contractility and cardiac output in *Zebrafish* (*Dash et al., 2017*), caused an alteration of myosin heavy-chain localization and disorganization of muscle fibers in somatic muscle (*Dash et al., 2017*), and significantly affected cytoskeleton dynamics in the ophthalmic artery (*Perumal et al., 2020*). To our knowledge, apart from these observations, no data is available on the function of Septin7 in striated skeletal muscle.

Here, we give evidence on the crucial role of Septin7 in skeletal muscle physiology using the C2C12 cell line and a Septin7 conditional knockdown mouse model established in our laboratory. We demonstrate an ontogenesis-dependent expression of Septin7, its effect on the phenotype and the in vivo and in vitro force generation. Furthermore, we provide evidence that Septin7 is essential in proper muscle development and myotube differentiation and demonstrate its presumed contribution to muscle regeneration.

## Results

### Skeletal muscle fibers express different septin isoforms

mRNA expression of representative members from all homology groups of septins has been detected in total lysates of human skeletal muscle by RT-PCR reactions on *m. quadriceps femoris* from amputated limbs (*Figure 1A*, *Figure 1—figure supplement 1A*). Most of the septin isoforms have been confirmed at the mRNA level in skeletal muscles originating from neonatal (4-day-old) C57BL/6J mice, as well. The expression pattern was similar in muscles of adult mice (4-month-old), although some of the isoforms (*Septin5-Septin10*) showed decreased intensity (*Figure 1B*). In parallel to the age-related alteration in mRNA expression of certain septin isoforms, a differentiation-dependent expression pattern of septins was revealed in C2C12 cells (*Figure 1—figure supplement 1B*). *Septin1* and *Septin3* were barely expressed, *Septin4* and *Septin7* expression showed gradual increase, while signals for other isoforms were rather uniform at different time points. *Septin12* and *Septin14* – similarly to human and mouse muscle – were not detectable.

### *Septin7* shows ontogenesis-dependent expression in skeletal muscle

Since the aforementioned alteration in its mRNA expression level during proliferation indicates a potential role in skeletal muscle development, Septin7 expression was investigated in detail. Varying expression of Septin7 has also been observed at the protein level during muscle development. Septin7 protein expression declined gradually with age, as evidenced by comparing muscle samples from 4-week-old and 4-month-old mice to newborns. On the other hand, *Septin7* expression seems to be independent of muscle type since signals were similar in different types of muscles (*m. tibialis anterior* – TA, *m. extensor digitorum longus* – EDL, and *m. soleus* – Sol) at the selected time points (*Figure 1C and D*). This ontogenesis-dependent decline in the Septin7 protein expression is consistent with our results at the mRNA level (*Figure 1—figure supplement 1C*). Septin7 expression is remarkably elevated in proliferating cell lines and newborn (NB) compared to differentiating cultures and adult muscle samples, respectively (*Figure 1—figure supplement 1D*).

### Localization of Septin7 on isolated skeletal muscle fibers

To assess the exact localization of Septin7 within the skeletal muscle, enzymatically isolated *m. flexor digitorum brevis* (FDB) fibers from BL6 mice were subjected to specific immunolabeling. As the position of skeletal muscle-specific α-actinin and RyR1 is well defined in muscle fibers, they served as points of reference in these experiments (*Figure 1E*). Septin7 was visualized by immunocytochemistry (*Figure 1F*) and merged images were generated with α-actinin and RyR1 to determine the relative localization of Septin7 (*Figure 1G*). Data acquired from confocal images showed that Septin7 is located next to the Z-line similarly to α-actinin and Septin7 signals were observed between the terminal cisternae characterized by RyR1 (*Figure 1H*).

### Skeletal muscle-specific downregulation of Septin7 results in an altered phenotype

To assess the function of Septin7 in skeletal muscle, a mouse model with skeletal muscle-specific knockdown of *Septin7* gene using the Cre/Lox system was generated (*Figure 2A*). Briefly, HSA-Cre transgenic mice have the *Cre* recombinase gene driven by the human alpha-skeletal actin (*ACTA1*) promoter. When bred with mice containing a *loxP*-flanked sequence of exon 4 encoding the GTP-binding P-loop of *Septin7* in the mouse genome, Cre-mediated recombination will catalyze exon 4-excision, resulting in an additional frame shift mutation downstream to this exon. Cre+ hemizygous mice following 3 months of tamoxifen feeding have been selected for the in vivo and in vitro experiments (*Figure 1—figure supplement 1E*). To estimate muscle-specific downregulation of Septin7 protein expression, total lysates of *m. quadriceps femoris* and *m. pectoralis* were examined. Partial deletion resulted in a significant and similar decrease of Septin7 protein expression in both muscle types studied (to 59 ± 8 and 63 ± 8% compared to Cre- mice, in *m. quadriceps* and *m. pectoralis*, respectively; *Figure 1—figure supplement 1F and G*) ,which correlated well with the approximately 50% deletion based on the three-primer PCR method of gDNA that was achieved in the different skeletal muscle types (*Figure 2B and C*). In addition, a pronounced spinal deformity was manifest in Cre+ mice in which the spine curved excessively outward, creating a hunchback (*Figure 2A*,

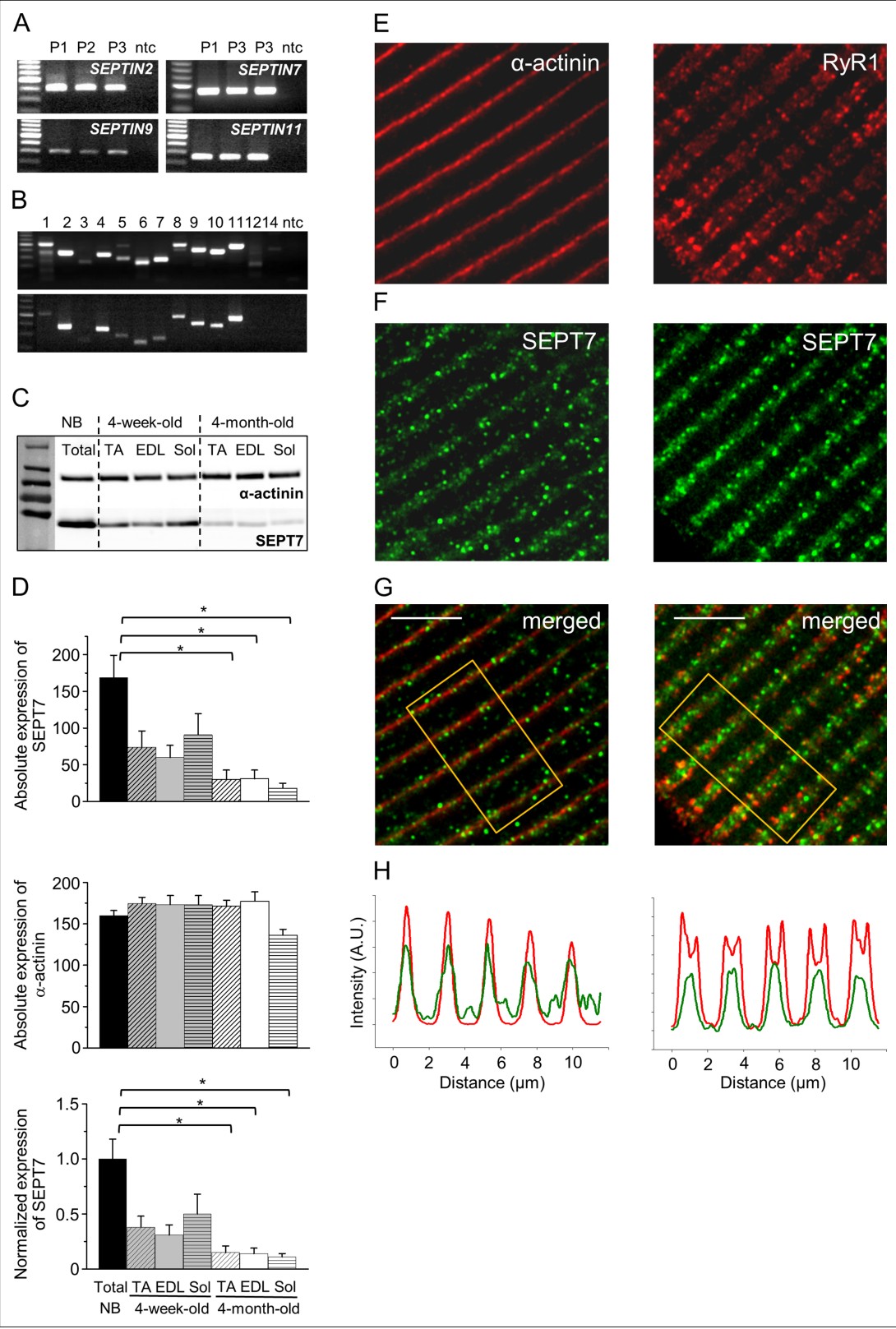

**Figure 1.** Septins are an integral part of the skeletal muscle cytoskeleton. (**A**) Agarose gel images showing the expression of septin isoforms at mRNA level in human skeletal muscle (*m. quadriceps femoris* from amputated limbs). Representative members from each homology group are shown. Independent samples from three patients were examined. Here and in all subsequent figures DNA Ladder is from Promega (G2101). First and last lines in the ladders correspond to 200 and 800 bp, respectively. (**B**) Expression of all septin isoforms at mRNA level in neonatal (upper panel) and adult

*Figure 1 continued on next page*

*Figure 1 continued*

(lower panel) mouse skeletal muscle. Non-template control (ntc) samples contain nuclease-free water instead of cDNA. First and last lines in the ladders correspond to 200 and 700 bp, respectively. (**C**) Differential expression of Septin7 (SEPT7, 50 kDa) at protein level during development in different types of skeletal muscle in mice, and α-actinin (110 kDa) was used as a normalizing control. Bar graphs represent the absolute expression of Septin7, α-actinin, and the normalized Septin7 expression in the different samples. Total skeletal muscle lysate of newborns (NB), TA: *m. tibialis anterior*; EDL: *m. extensor digitorum longus; and* Sol: *m. soleus.* Here and all subsequent figure Page Ruler Plus Prestained Protein Ladder from Thermo Fisher (26620) was used. First and last lines in the ladder correspond to 55 and 250 kDa, respectively. (**D**) Pooled data of absolute expression of Septin7, absolute expression of actin, and normalized Septin7 expression during development. Representative data of three mice/age group. Data represent mean ± standard error of the mean (SEM), *p<0.05 from ANOVA. Intracellular localization of Septin7 (**F**) in adult skeletal muscle relative to α-actinin, and RyR1 (**E**) using immunofluorescence staining, and the merged images for the aforementioned proteins (**G**). Scale bar is 5 μm. (**H**) Fluorescence intensity changes of Septin7 (green) and α-actinin or RyR1 (red) along the fiber calculated from the rectangular area in panel (**G**). See also *Figure 1—figure supplement 1*.

The online version of this article includes the following source data and figure supplement(s) for figure 1:

**Source data 1.** Agarose gel images showing the expression of *Septin1-4* isoforms at mRNA level in human skeletal muscle (*m. quadriceps femoris* from amputated limbs).

**Source data 2.** Agarose gel showing the expression of *Septin5-8* isoforms at mRNA level in human skeletal muscle.

**Source data 3.** Agarose gel showing the expression of *Septin9-12* isoforms at mRNA level in human skeletal muscle.

**Source data 4.** Expression of all septin isoforms at mRNA level in neonatal mouse skeletal muscle.

**Source data 5.** Expression of all septin isoforms at mRNA level in adult mouse skeletal muscle.

**Source data 6.** Differential expression of Septin7 (50 kDa) at protein level during development in different types of skeletal muscle in mice represented without protein molecular weight.

**Source data 7.** Differential expression of Septin7 (50kDa) at protein level during development in different types of skeletal muscle in mice represented with protein molecular weight.

**Figure supplement 1.** Expression of different septin isoforms at mRNA and protein level in C2C12 cell culture, mouse and human skeletal muscle, highlighted genetic modification of Septin7 expression.

**Figure supplement 1—source data 1.** Expression of *SEPTIN1-4* isoforms at mRNA level in human skeletal muscle (*m. quadriceps femoris*).

**Figure supplement 1—source data 2.** Expression of *SEPTIN5-8* isoforms at mRNA level in human skeletal muscle.

**Figure supplement 1—source data 3.** Expression of *SEPTIN9,10,11,14* isoforms at mRNA level in human skeletal muscle.

**Figure supplement 1—source data 4.** mRNA expression of *Septin1-8* isoforms in C2C12 cultured cells.

**Figure supplement 1—source data 5.** mRNA expression of *Septin9-12* and *Septin14* isoforms in C2C12 cultured cells.

**Figure supplement 1—source data 6.** Age-related (age of 4 days [4d], 4 weeks [4w], 4 months [4m]) mRNA expression of *Gapdh* in newborn and in different skeletal muscle types (*m. flexor digitorum brevis, m. extensor digitorum longus*, and *m. soleus*) of adultmouse.

**Figure supplement 1—source data 7.** Age-related mRNA expression of *Gapdh* in newborn and in *m. tibialis anterior* skeletal muscle type of adult mouse.

**Figure supplement 1—source data 8.** Age-related mRNA expression of *Septin7* in newborn and in different skeletal muscle types (newborn, *m. flexor digitorum brevis, m. externsor digitorum longus,* and *m. soleus*) of adult mouse.

**Figure supplement 1—source data 9.** Age-related mRNA expression of *Septin7* in newborn and in *m. tibialis anterior* skeletal muscle type of adult mouse.

**Figure supplement 1—source data 10.** Ontogenesis-dependent Septin7 (SEPT7) protein expression in proliferating and differentiated C2C12 cells, newborn and different muscle types of adult (4 months old) mice; α-actinin was used as control.

**Figure supplement 1—source data 11.** Screening for the presence of the *HSA-MCM* transgene from genomic DNA by PCR in Cre+ and Cre- mice.

**Figure supplement 1—source data 12.** α-Actinin protein expression in Cre+ mice following a tamoxifen diet compared to Cre- and control BL6 mice.

**Figure supplement 1—source data 13.** Altered Septin7 protein expression in Cre+ mice following a tamoxifen diet compared to Cre- and control Bl6 mice.

*Figure 2—figure supplement 1A and B*). From CT images of three individual Cre+ and Cre- mice, we determined the average angle of the 10th thoracic vertebra (*Figure 2—figure supplement 1C*), and this parameter was significantly smaller in Cre+ animals compared to the Cre- littermates, providing further evidence for the deformity of the vertebra (*Figure 2—figure supplement 1D*). As tamoxifen alone did not induce this deformity, it is likely to be the consequence of muscle atrophy and a weaker muscle tone. In line with the visible morphological changes, Cre+ mice had significantly smaller body weight starting from the age of the seventh week (*Figure 2D*). The effect of tamoxifen treatment can be excluded since no significant difference was found between control BL6 and Cre- mice in terms of phenotype and the body weight gain (*Figure 2—figure supplement 2A and B*, respectively).

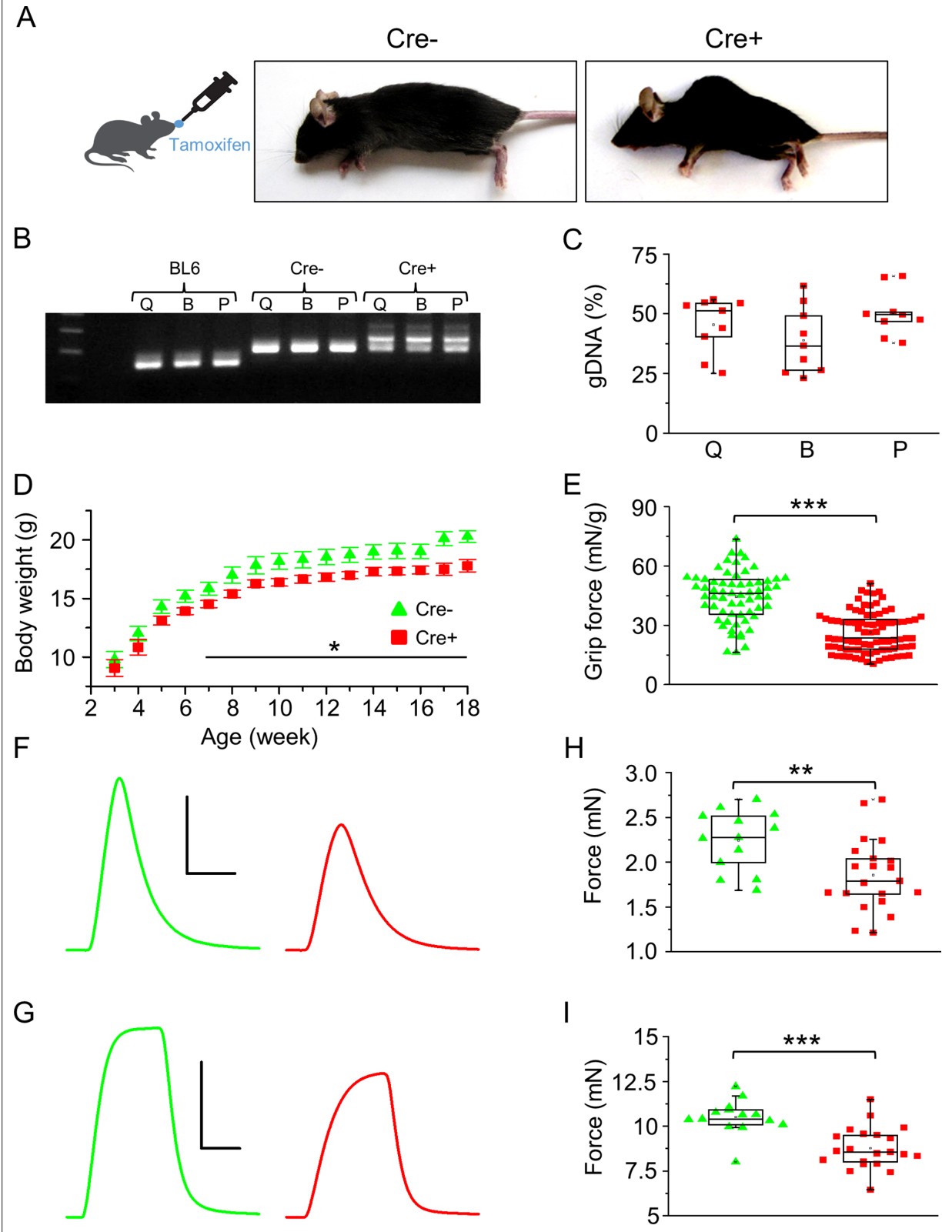

**Figure 2.** Skeletal muscle-specific knockdown of Septin7 resulted in a severe phenotype. (**A**) Images of tamoxifen-fed Cre- and Cre+ mice (both *Septin7^flox/flox*) at the age of 4 months (see *Figure 2—figure supplement 1*). (**B**) Three-primer PCR for detecting the partial deletion of the *Septin7* gene in different skeletal muscle types of Cre+ mice (Q: *m. quadriceps femoris*; B: *m. biceps femoris*; P: *m. pectoralis*) and the lack of deletion in samples prepared from BL6 control and Cre- mice. In samples originated from Cre- animals, the floxed exon 4, while in wild-type tamoxifen-fed mice the

*Figure 2 continued on next page*

*Figure 2 continued*

unmodified exon 4 is demonstrated. First and last lines in the ladder correspond to 200 and 300 bp, respectively. (**C**) Pooled data of the percentage of exon 4 deletion in different muscle types of Cre+ mice. 14 littermates (nine Cre+ and five Cre-) were examined from three litters. Here and in all subsequent figures, the rectangles in the box plots present the median and the 25 and 75 percentile values, while the error bars point to 1 and 99%. (**D**) Changes of body weight in Cre- (green triangle, n = 11) and floxed Cre+ (red square, n = 14) mice. Black solid line shows where the difference is statistically different (p<0.05 from t-test). Error bars show SEM. (**E**) Grip force normalized to body weight in Cre- (n = 4) and Cre+ mice (n = 7). Representative twitch (**F**) and tetanic force (**G**) transients in *m. extensor digitorum longus* (EDL). Peak twitch (**H**) and tetanic force (**I**) in EDL from Cre- (n = 7) and Cre+ (n = 11) mice. Calibration in panel (**F**): 1 mN and 50 ms; (**G**): 5 mN and 100 ms. \*\*p<0.01, \*\*\*p<0.001 (from t-test). See also ***Figure 2—figure supplement 2***. Main contractile proteins (actin and myosin) and L-type calcium channel distribution within single *m. flexor digitorum brevis* (FDB) myofibers isolated from Cre- and Cre+ animals were also investigated (see ***Figure 2—figure supplement 3***).

The online version of this article includes the following source data and figure supplement(s) for figure 2:

**Source data 1.** Three-primer PCR was used to detect the partial deletion of the *Septin7* gene in different skeletal muscle types of Cre+ mice (Q: *m. quadriceps femoris*; B: *m. biceps femoris*; P: *m. pectoralis*) and the lack of deletion in samples prepared from BL6 control and Cre- mice.

**Figure supplement 1.** Effects of Septin7 knockdown on the phenotype of Cre+ mice.

**Figure supplement 2.** Effects of tamoxifen feeding on the phenotype of BL6 and Cre- mice.

**Figure supplement 3.** Expression and spatial distribution of actin, MYH4, and L-type calcium channels in single *m. flexor digitorum brevis* (FDB) muscle fibers isolated from Cre- and Cre+ animals.

## In vivo physical performance is impaired in Septin7 knockdown mice

In vivo experiments have led to concordant observations that smaller body weight was accompanied by an impaired muscle performance. The average grip force normalized to body weight was similar in control BL6 and Cre- mice (***Figure 2—figure supplement 2C***), while significantly smaller values were measured in Cre+ animals (***Figure 2E***). Voluntary running tests provided identical results. All parameters of running (distance, duration, average speed, maximal speed) were significantly smaller in Cre+ animals compared either to BL6 or Cre- mice, while there was no difference between the parameters of running in BL6 and Cre- mice (***Supplementary file 1b***).

## In vitro force is reduced in Septin7 knockdown animals

The significant influence of Septin7 reduction on muscle force parameters measured in vivo was also reflected in in vitro experiments. Both twitch and tetanic force decreased significantly in EDL (***Figure 2F–I***) and Sol muscle (***Figure 2—figure supplement 2H–K***) of Cre+ mice alike compared to the same muscles of Cre- littermates. This reduction in maximal force was the result of reduced Septin7 content since tamoxifen treatment alone had no effect on the contractile parameters (***Figure 2—figure supplement 2D–K***). Interestingly, the kinetics of twitches and tetani as well as the fatigability changed only in Sol of Cre+ mice (***Supplementary file 1c***). In vivo and in vitro force measurements thus suggest that Septin7 fundamentally contributes to the normal skeletal muscle performance.

Furthermore, the expression and spatial distribution of the two contractile proteins, actin (***Figure 2—figure supplement 3A***) and myosin (MYH4) (***Figure 2—figure supplement 3B***), on single muscle fibers isolated from FDB of Cre- and Cre+ animals were examined. Significant alteration of the contractile proteins was not detected; in addition, the expression pattern of L-type calcium channel (***Figure 2—figure supplement 3C***) was also similar when the images of Cre- and Cre+ samples were compared.

## Septin7 is critical for proper cellular development and myotube differentiation

In proliferating C2C12 myoblasts, the cellular distribution of Septin7 (green) was detected along with the actin-network (red) using immunocytochemistry (***Figure 3A***), which revealed filamentous structure of Septin7 and its co-localization with actin.

The functional role of Septin7 in C2C12 cells was evaluated using the CRISPR/Cas9 technique in control cells (***Figure 3B***). This approach led to abnormal cell size (***Figure 3C***). In addition, cell proliferation was stopped, providing an additional clue for the importance of Septin7 in cell division. Because cell proliferation was too severely affected under this condition, the milder, partial knockdown approach using shRNA was applied to assess the impact of Septin7 on myoblast proliferation and myotube formation. The suppressed protein level of knockdown (S7-KD) cells (***Figure 3D***) has

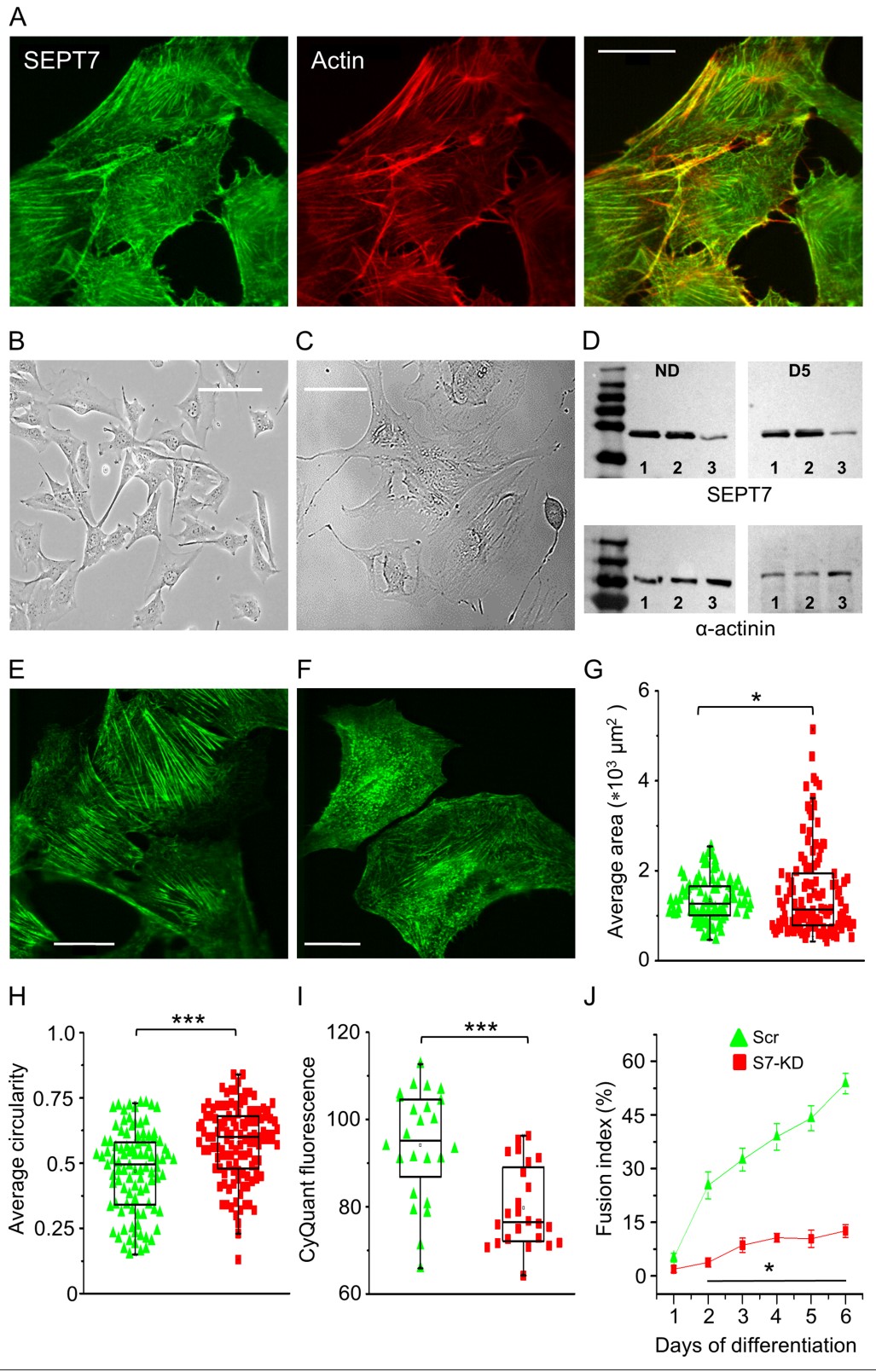

**Figure 3.** Septin7 is critical for proper cellular development and myotube differentiation. (**A**) Confocal images of Septin7 immunolabeling (green) and actin filaments (red) and their co-localization in control C2C12 cells. Calibration is 20 µm. Transmitted light images of control (**B**) and Septin7 knockout (KO; **C**) cells demonstrating that complete KO of Septin7 in C2C12 cells prevents appropriate cell proliferation. Scale bar is 50 µm for both the

*Figure 3 continued on next page*

*Figure 3 continued*

control and KO cells. (**D**) Partial knockdown (KD) of Septin7 expression at proliferation stage (nondifferentiated [ND]) and 5 days after differentiation induction (D5). 8 µg of protein samples from absolute control (Ctrl), scrambled transfected (Scr), and S7-KD cells in each case were loaded to SDS-PAGE gel, and following electrophoresis and blot transfer into nitrocellulose membrane, Septin7 (50 kDa) and α-actinin (110 kDa) were probed with the appropriate primary antibodies. Numbers 1, 2, and 3 at the bottom of the gels indicate Ctrl, Scr, and S7-KD samples, respectively. First and last lines in the ladders correspond to 35 and 250, and 70 and 250 kDa, for the upper and lower panels, respectively. Immunolabeling of Septin7 filaments in Scr (**E**) and S7-KD cells (**F**) demonstrating altered filament structure, cell size, and shape. Scale bar is 20 µm. Quantification of changes in cell morphology, area (**G**) and circularity (**H**) in S7-KD cultures. Green triangles represent Scr, while red squares represent S7-KD cells. The number of cells investigated was 96 in Scr and 121 in S7-KD cultures; *$p<0.05$, ***$p<0.001$ from t-test; experiment was repeated twice (N = 2). (**I**) Decreased proliferation 3 days after the plating assessed as CyQUANT fluorescence (n = 24; N = 3) and (**J**) suppressed differentiation of S7-KD cells determined by the calculation of fusion index during 6 days of investigation (n = 20; N = 2). Horizontal line in (**J**) shows where the difference between Scr and S7-KD cells was statistically significant (*$p<0.05$ from t-test). See also *Figure 3—figure supplement 1* and *Figure 3—figure supplement 2*.

The online version of this article includes the following source data and figure supplement(s) for figure 3:

**Source data 1.** Partial knockdown (KD) of Septin7 protein expression at proliferation stage (nondifferentiated [ND]) and following differentiation induction (D1–D2).

**Source data 2.** Partial knockdown (KD) of Septin7 protein expression following differentiation induction (D3-D6).

**Source data 3.** α-actinin protein expression at proliferation stage (ND) and following differentiation induction (D1-D2) in cultured C2C12 cells.

**Source data 4.** α-actinin protein expression following differentiation induction (D3-D6) of cultured C2C12 cells.

**Figure supplement 1.** Septin7 filament structure and the effect of gene silencing on cellular parameters.

**Figure supplement 1—source data 1.** Modified expression of Septin7 protein (50 kDa) in non-differentiated stage and 1-2 days following differentiation induction of Ctrl, Scr, and S7-KD cells.

**Figure supplement 1—source data 2.** Modified expression of Septin7 protein during the differentiation (D3-D6) program of Ctrl, Scr, and S7-KD cells.

**Figure supplement 2.** SEPT 7 downregulation modifies the intracellular architecture of C2C12 cells.

been preserved along the differentiation process (*Figure 3—figure supplement 1D*) compared to absolute control or scrambled shRNA-transfected (Scr) cells. Having repeated the immunolabeling in Scr (*Figure 3E*) and S7-KD cells (*Figure 3F*), strongly modified cell morphology and cytoplasmic distribution of Septin7 was revealed. The characteristic filamentous structure of Septin7 disappeared in most of the S7-KD cells; instead, a fragmented, pointwise appearance of the protein was observed. In control and Scr cultures, Septin7 and actin filaments were highly co-localized, while in S7-KD cells this spatial overlap was disrupted, although the actin structure within the cells seemed mostly unchanged (*Figure 3—figure supplement 1A–C*).

Changes in cell morphology were further analyzed, and significant differences in average cell cross-sectional area and circularity were found in S7-KD cells compared to Scr cultures (*Figure 3G and H*). These data express quantitatively the conspicuous visual observations, that is, in S7-KD cultures a specific population of cells was present, which were large in size and lost their processes, resulting in a more rounded shape. As revealed by the individual data points presented in *Figure 3G*, a portion of KD cells have approximately the same size as the Scr ones; however, a large number of cells in KD cultures have significantly increased area. These types of cells were not observed in control or Scr cultures. At the same time, control and Scr cells did not show any significant changes with respect to these parameters (*Figure 3—figure supplement 1E*), while the perimeter of the individual cells revealed no significant difference between control, Scr, and KD cultures (*Figure 3—figure supplement 1F*). Intracellular architecture of Septin7 and actin filaments in Scr cells (*Figure 3—figure supplement 2A and B*) and its prominent changes in S7-KD cultured cells (*Figure 3—figure supplement 2C and D*) has been further proven by masking confocal images. Not only the significant changes in cell morphology were evident between the two cell types, but the remarkable alteration of filamentous structure was also well defined.

Myoblast proliferation capacity was also tested and significantly reduced proliferation was observed in Septin7-modified cells (*Figure 3I*). Myotube formation was assessed by calculating the

fusion index, the degree to which myogenic nuclei were found in multinucleated, terminally differentiated myotubes. From the second day of differentiation onward, increasing myotube formation was observed in control and Scr cultures (*Figure 3—figure supplement 1G*), while only negligible cell fusion was detected in S7-KD cultures, even at the sixth day of the experiment (*Figure 3J*, *Figure 3—figure supplement 1G*).

## In vivo knockdown of Septin7 alters myofibrillar structure and mitochondrial parameters

Structural changes within the myofibrillar system of adult skeletal muscle fibers triggered by the reduced Septin7 expression were studied using electron microscopy (EM) on TA muscle. Representative EM images taken on transversal muscle sections of Cre- animals (*Figure 4A*) demonstrated the presence of myofibrils of a single fiber, which are all well demarcated with a visible SR. In samples of Cre+ mice (*Figure 4B*), however, separation of myofibrils was less obvious. After identifying all individual myofibrils within an area of the actual visual field, average area, perimeter of the individual myofibrils, and the average number of myofibrils have been estimated within a given unit of area (1 μm²) of the actual visual field. As *Figure 4C* demonstrates, both the average area and the perimeter of myofibrils from Cre+ animals were significantly smaller than the corresponding parameters of myofibrils from Cre- mice. In line with the above, the number of myofibrils within a given area of the visual field (*Figure 4D*) was increased significantly in samples originating from Cre+ mice compared to sections of Cre- animals. In myofibrils from Cre+ mice, all the aforementioned parameters were also significantly different from the control BL6 samples. On the other hand, there was no significant difference between the corresponding parameters of myofibrils isolated from BL6 and Cre- animals (*Figure 4—figure supplement 1C and D*), suggesting that the markedly reduced size and increased number of the individual myofibrils per unit area in Cre+ mice are attributable to the modified Septin7 expression in their skeletal muscle.

The aforementioned parameters have also been calculated for the mitochondria using similar transversal sections of TA muscles dissected from the different animal groups. In *Figure 5A and B*, representative images on muscle sections from Cre- and Cre+ mice are presented, respectively. To describe the changes in morphology, the area, perimeter, aspect ratio (AR), and form factor (FF) were calculated for each identified mitochondria in all EM micrographs. The calculated average perimeter (*Figure 5C*) and area (*Figure 5D*, inset) were found to be significantly increased in Cre+ samples compared to data of either muscles from Cre- animals or to the samples of control BL6 mice. The relative distribution of the mitochondrial area was also slightly altered in Cre+ samples (*Figure 5D*) with the appearance of larger mitochondria, AR was significantly reduced in Cre+ samples compared to Cre-, while there was no significant difference in the calculated FF parameters of the two sample groups (*Figure 5C*). The number of mitochondria per unit area within the selected visual fields was also significantly increased in samples from Cre+ mice compared with the muscles of Cre- or BL6 mice, as shown in *Figure 5E*. Mitochondria-related parameters (*Figure 5—figure supplement 1C and D*) were not different between the BL6 control and Cre- groups. To illustrate better the lack of change in the area and perimeter of myofibrils/mitochondria in Cre- and BL6 control samples, statistical distribution of the datasets was calculated and plotted as histograms. As *Figure 4—figure supplement 1A and B* and *Figure 5—figure supplement 1A and B* represent, area and perimeter of myofibrils and mitochondria, respectively, show no or very slight shift between samples from control and Cre- mice.

Transmission electron microscopic analysis of longitudinal sections from the different animal groups revealed normal myofibrillar structure (sarcomere length, triad composition) in Cre- samples, while the occurrence of large mitochondrial networks has been identified in most images taken from Cre+ muscles, as demonstrated by representative images in *Figure 5F and G*, respectively. Assessing the morphological parameters revealed clear changes in the average area (*Figure 5I*, inset), perimeter, AR, and FF (*Figure 5H*) in longitudinal sections. These parameters were significantly higher in Cre+ samples compared with the Cre- counterparts. The relative distribution of mitochondrial area demonstrates the appearance of mitochondria with large areas (*Figure 5I*) that were not present in Cre- mice. The total area of mitochondria as a percentage of the total area of the corresponding visual field was also determined in BL6 and Cre- animals with no significant difference in this parameter between these mice (*Figure 5—figure supplement 1E*).

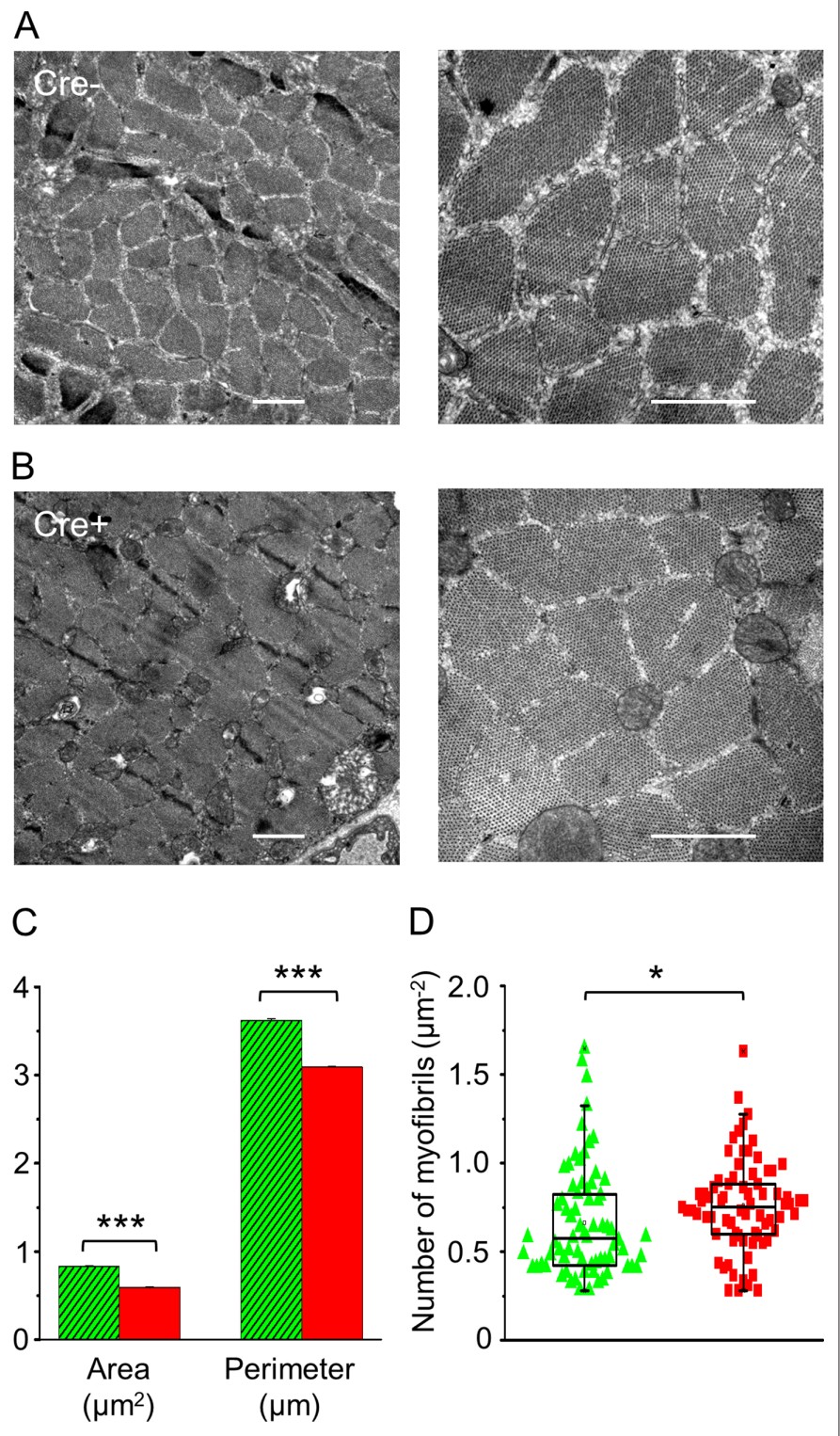

**Figure 4.** In vivo knockdown of Septin7 alters myofibrillar structure. (**A, B**) Representative electron microscopy (EM) images of myofibrils from cross-sectional samples of *m. tibialis anterior* (TA) muscles from Cre- and Cre+ mice at smaller (left) and larger (right) magnification where scale bars represent 1 μm. (**C, D**) Area and perimeter of the individual myofibrils and the number of myofibrils within 1 μm$^2$ of an appropriate visual field were determined from EM images. Here and in all subsequent figures, green columns represent data from Cre-, while red columns from Cre+ animals (average ± SEM). The total number of myofibrils counted for area and perimeter was 3012 and 3174

*Figure 4 continued on next page*

*Figure 4 continued*

in Cre- and Cre+ mice, respectively, while the number of visual fields examined was 72 and 74, for calculating the number of myofibrils (*p<0.05; ***p<0.001 from t-test). See also *Figure 4—figure supplement 1*.

The online version of this article includes the following figure supplement(s) for figure 4:

**Figure supplement 1.** Changes in myofibrillar parameters with in vivo knockdown of Septin7.

Mitochondrial DNA content in different muscles, *m. pectoralis* and *m. quadriceps,* was also determined. Mitochondrial (16S) RNA content was found to be severely reduced in muscles from Cre+ mice compared to data of Cre- mice in both examined muscles, respectively (*Figure 5J*). These results suggest that not only mitochondrial morphology but presumably mitochondrial function is also severely affected by the alteration of Septin7 expression in skeletal muscles.

In addition, changes in mitochondrial morphology were also examined in cultured C2C12 cells. In Septin7-modified KD cells (*Figure 5—figure supplement 2C*), not only the filamentous structure of Septin7 was different compared to the control (*Figure 5—figure supplement 2A*) and Scr (*Figure 5—figure supplement 2B*) cultures, but the mitochondrial network was also severely affected following the Septin7 gene silencing.

## Septin7 expression is modified during skeletal muscle regeneration

Previous data (shown in *Figure 1C and D* and *Figure 1—figure supplement 1C and D*) suggested that Septin7 has a prominent role in muscle development and structural assembly of newborn animals and in proliferating C2C12 cells, thus changes of its expression during muscle regeneration were analyzed. Mild muscle injury in TA muscles was induced by in vivo BaCl$_2$ injection in mice. At two different time points, cryosections were prepared from the injected, and the contralateral, noninjected control muscles and were subjected to hematoxylin-eosin (HE) staining. *Figure 6—figure supplement 1* presents images from noninjected (A) and injected muscles (B) (as indicated) 14 days after the treatment. Satellite cell-coupled Pax7 expression was observed during regeneration. More intensive PAX7-positive signal and regeneration-induced, central nuclei in the myofibrils were detected in injected muscles (see magnified region of *Figure 6—figure supplement 1A and B*) in *Figure 6B* compared to noninjected controls (*Figure 6A*) using DAB reaction. As demonstrated in *Figure 6D and F*, the level of Pax7 expression was found to be significantly higher in the injected than in the noninjected muscles. Septin7 protein expression was also monitored for 2 weeks during regeneration (*Figure 6C*), and Western blot analysis revealed pronounced overexpression at each time point investigated (*Figure 6E*). Similar upregulation of Septin7 and Pax7 detected upon BaCl$_2$-induced muscle injury suggests that cytoplasmic septins potentially contribute to regeneration via regulating proliferation and differentiation of newly formed muscle cells.

## Discussion

### Septin filaments are an integral part of the skeletal muscle cytoskeleton

Specific members of different septin isoforms are present from yeasts to mammals during the phylogenesis (see review *Gönczi et al., 2021*). They have been reported to interact with other cytoskeletal elements, with the plasma membrane, and furthermore, to be involved in a number of signaling pathways (*Xie et al., 1999*; *Sisson et al., 2000*; *Joberty et al., 2001*; *Kinoshita et al., 1997*; *Surka et al., 2002*). In this study, we demonstrate for the first time the presence of different septin isoforms and the contribution of Septin7 to skeletal muscle physiology.

### Expression of septin isoforms in skeletal muscle

As septins are known to be the fourth component of the cytoskeleton, they could participate in the skeletal muscle architecture and function as well. Based on their mRNA expression profile in mouse and human skeletal muscle samples, the presence of at least one essential member of each homology group indicates the possibility of build-up of the previously described hetero-oligomeric structure (*Sirajuddin et al., 2007*; *Jiao et al., 2020*; *DeRose et al., 2020*; *Soroor et al., 2021*; *Mendonça*

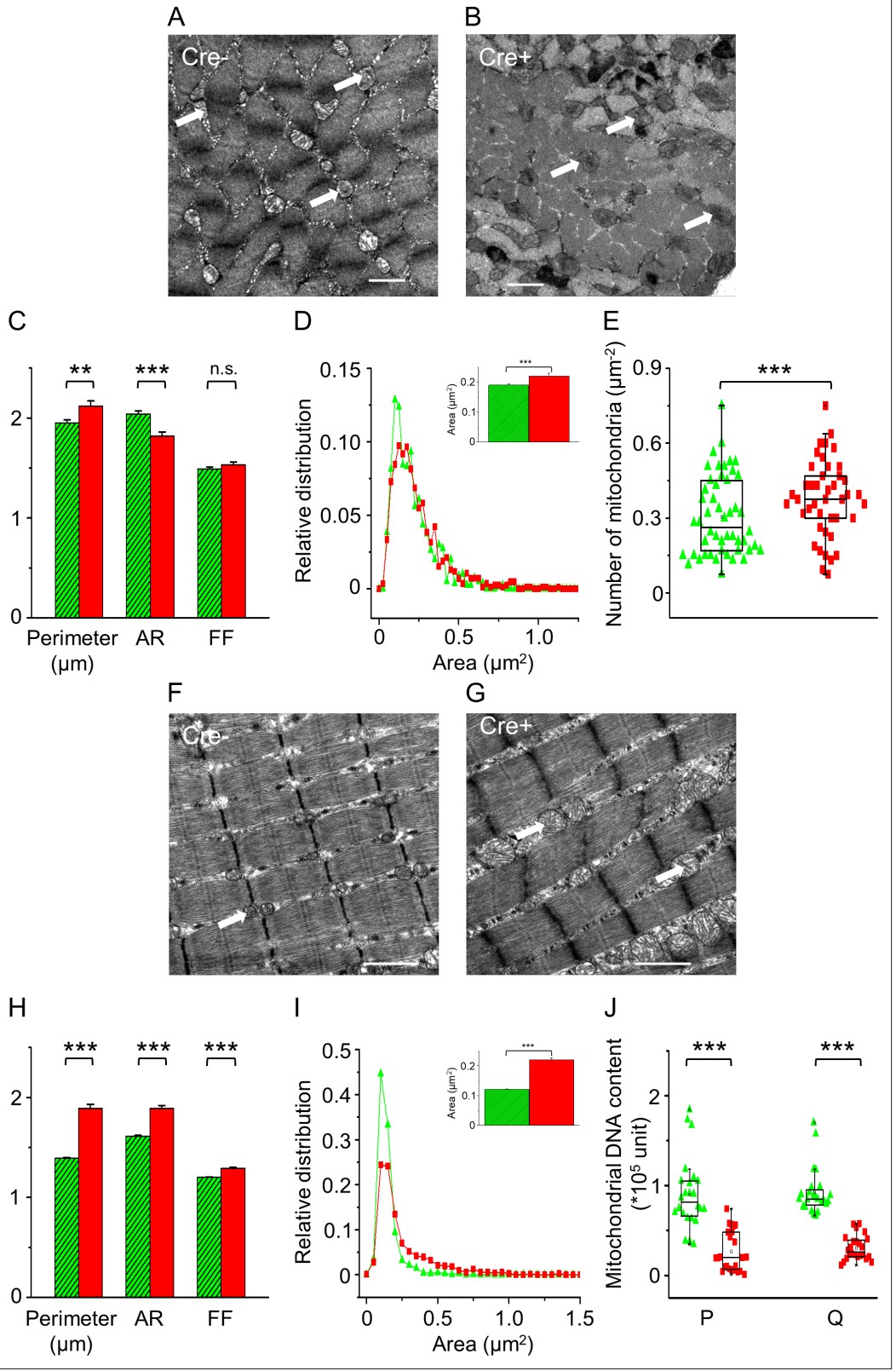

**Figure 5.** In vivo knockdown of Septin7 alters mitochondrial parameters. (**A, B**) Representative electron microscopy (EM) images of myofibrils with mitochondria from cross-sectional samples of *m. tibialis anterior* (TA) muscles of Cre- and Cre+ mice. (**C**) Perimeter, aspect ratio (AR), and form factor (FF) of the individual mitochondria (average ± SEM; **p<0.01, ***p<0.001 from t-test). (**D**) Statistical distribution of mitochondrial area determined

*Figure 5 continued on next page*

*Figure 5 continued*

from EM images of Cre- (green) and Cre+ (red) samples. The average area with SE is presented in the inset of this panel. (**E**) The number of mitochondria within 1 µm$^2$ of an appropriate visual field was calculated from EM images. The total number of mitochondria counted for area, perimeter, AR, and FF was 811 and 866 in Cre- and Cre+, respectively, while the number of visual fields examined was 50 and 43 for calculating the number of mitochondria (***p<0.001 from t-test). (**F, G**) Representative EM images of myofibrils and mitochondria from longitudinal sections of TA muscle samples originating from Cre- and Cre+ mice. (**H**) Perimeter, AR, and FF were determined from the longitudinal sections in both Cre- (n = 1635) and Cre+ (n = 713) samples (average ± SEM; ***p<0.001 from t-test). (**I**) The relative distribution of mitochondrial area, average area of mitochondria in Cre- and Cre+ skeletal muscle fibers is presented in the inset of this panel. The number of visual fields investigated was 42 and 41, respectively (average ± SEM; ***p<0.001 from t-test). Scale bars are equal to 1 µm in all images. (**J**) Mitochondrial DNA content was determined from *m. pectoralis* (P) and *m. quadriceps femoris* (Q) of Cre- and Cre+ mice using specific qPCR primers. The number of samples was 8, while in each case experimental triplicate was performed (***p<0.001 from t-test). See also *Figure 5—figure supplement 1*. Changes in mitochondrial morphology following Septin7 depletion were also studied in cultured C2C12 cells. For these results, see *Figure 5—figure supplement 2*.

The online version of this article includes the following figure supplement(s) for figure 5:

**Figure supplement 1.** Changes in mitochondrial parameters with in vivo knockdown of Septin7.

**Figure supplement 2.** Alteration of mitochondrial network and septin filaments following Septin7 deletion in C2C12 cultures.

*et al., 2021*). These results predict the presence of a higher-order oligomeric structure of septins, though further structural investigations are needed.

Septin7 showed a differential, declining expression with age both in fast and slow twitch skeletal muscles, which may emphasize its significance in muscle development. In line with the observation, that genetic deletion of *Septin7*, *Septin9*, or *Septin11* resulted in embryonic lethality, reflecting their essential role in early, intrauterine development (*Hall et al., 2008*; *Füchtbauer et al., 2011*; *Röseler et al., 2011*). Considering that Septin7 is the only member of the SEPTIN7 homology group, its loss is not expected to be compensated for in the oligomers. Furthermore, the conditional Septin7 KD (Cre+) mice showed altered phenotype and produced decreased muscle force. This raises two questions: (1) are septins essential in the early development of skeletal muscle, and (2) do they lose their significance with aging?

Connection between septins and other cytoskeletal proteins has a prominent role in the conversion of mechanical inputs into biochemical signals. Septins have been shown to co-localize with actin filaments within stress fibers associated with focal adhesion complexes, as well as perinuclear actin. Septins participate in mechano-transduction by promoting the formation of contractile actomyosin networks, and the recruitment of myosin to actin in cancer-associated fibroblasts, in mammalian epithelial cells, and mouse cardiac epithelial cells (see the review *Lam and Calvo, 2019*). In accordance with the above, we have demonstrated the co-localization of Septin7 and actin filaments in control C2C12 cells, and furthermore, without Septin7 these cells were not able to proliferate. Knocking down Septin7 in C2C12 cells resulted in altered filament structure, cell size, and shape as well. This is in line with the observation that during embryogenesis interaction of Septin-2, 6, 7, and 9 with the cytosolic elements has an essential role in the regulation of cardiac functions (*Ahuja et al., 2006*).

## Suppression of septin expression in skeletal muscle results in severe skeletal deformities

Although septins were considered as key players of cell division based on their evolutionary conserved function in cytokinesis, they are now associated with a variety of other processes. Downregulation of the ubiquitous Septin7 has been implicated in several pathological events, including abnormal morphology and immature sperm (*Chao et al., 2010*); altered glucose uptake in the insulin-sensitive podocytes (*Wasik et al., 2012*); inefficient microvascular angiogenesis, actomyosin organizations, and directional migration in primary mouse cardiac endothelial cells (*Liu et al., 2014*). Septin7 has been suggested as a novel regulator of neuronal calcium homeostasis since in resting neurons suppressed dSeptin7 (homologue of human *SEPTIN7* in *Drosophila*) resulted in Ca$^{2+}$ store-independent opening of Orai, a calcium release-activated calcium channel (*Deb et al., 2016*). Also, Septin7 deficiency alters the morphology of mature neurons (*Tada et al., 2007*; *Xie et al., 2007*; *Cho et al., 2011*; *Yadav et al.,*

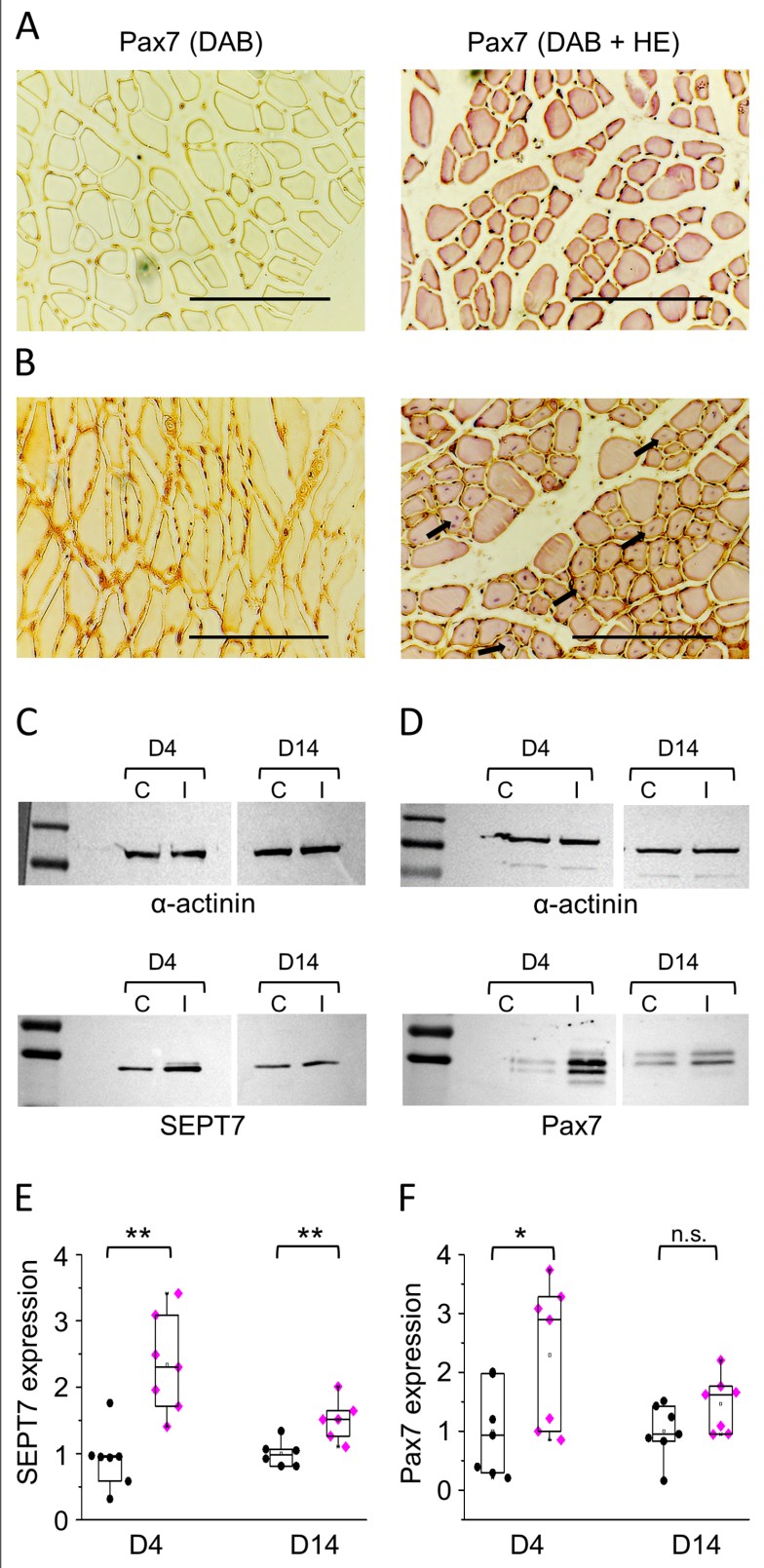

**Figure 6.** Septin7 and muscle regeneration. Representative histological images from noninjected (**A**) and BaCl₂-injected (**B**) *m. tibialis anterior* (TA) muscles of BL6 mice 14 days following the muscle injury. The 6 μm cryosections were subjected to Pax7-specific immunostaining either alone (left panels) or together with HE-staining (right panels); the latter represents regenerating myofibrils with central nuclei in BaCl₂-injected samples (some indicated

*Figure 6 continued on next page*

*Figure 6 continued*

by arrows). Scale bars are equal to 100 µm. (**C, D**) Protein samples were prepared from control (**C**) and injected (**I**) muscles at appropriate time points (day 4 [D4] and day 14 [D14]). Septin7 and Pax7 protein expression was determined in each sample pair, and α-actinin was used as normalizing control. First and last lines in the ladders correspond to 100 and 130, and 55 and 70 kDa in panel C, while 70 and 130, and 55 and 70 kDa in panel D, for upper and lower panels, respectively. Normalized Septin7 (**E**) and Pax7 (**F**) expression level was determined in control (black circles) and injected muscles (magenta diamonds), and plotted individually at each time point of investigation. Each point represents individual data (*p<0.05, **p<0.01 from t-test). See also *Figure 6—figure supplement 1*.

The online version of this article includes the following source data and figure supplement(s) for figure 6:

**Source data 1.** α-actinin protein expression incontrol (C) and injected (I) *m. tibialis anterior* (TA) muscle samples at appropriate time points (day 4 [D4] and day 14 [D14]).

**Source data 2.** Septin7 protein expression in control (C) and injected (I) *m. tibialis anterior* (TA) muscle samples at appropriate time points (D4 and D14).

**Source data 3.** α-actinin protein expression in samples prepared from control (C) and injected (I) *m. tibialis anterior* (TA) muscles at appropriate time points (day 4 [D4] and day 14 [D14]).

**Source data 4.** Pax7 protein expression in samples prepared from control (C) and injected (I) *m.tibialis anterior* muscles at appropriate time points (D4 and D14).

**Figure supplement 1.** Involvement of Pax7 in muscle regeneration.

*2017*), leading to neurological disorders such as Alzheimer's disease, schizophrenia, and neuropsychiatric lupus erythematosus.

As an ontogenesis-dependent expression of *Septin7* was found – largest expression in neonatal animals that declined until the age of 4 months (*Figure 1*, *Figure 1—figure supplement 1*) and then remained constant until 18 months (not shown) – interventions to alter its expression at different age of the animals were also examined. Unfortunately, the earliest possible application of tamoxifen, that is, administering it to pregnant Cre- females, resulted in stillborn offspring. Short applications, 1 month or less, in young or adult animals resulted in only a minor (<20%) reduction in Septin7 expression with inconclusive modifications in muscle function. These trials led us to use longer (3-month-long, as described) tamoxifen treatment, which gave appreciable (~50%) reduction at the protein level and definite changes in function. As we were concerned whether such long tamoxifen treatment might have effects on its own, tamoxifen-treated littermate Cre- mice were also compared to nontreated control animals. These experiments (*Figure 2—figure supplement 2*, *Figure 4—figure supplement 1*, *Figure 5—figure supplement 1*) confirmed that it was not tamoxifen feeding alone rather the consequent reduction in Septin7 expression that was responsible for the alterations seen in Cre+ mice.

In Septin7-modified Cre+ mice, generated in our laboratory, spinal structural deformity, similar to the human Scheuermann's kyphosis (*Sardar et al., 2019*) affecting the thoracic or thoracolumbar spine, was observed. The muscle atrophy and decreased muscle tone observed in Cre+ animals are assumed to be responsible for this vertebral defect since tamoxifen treatment alone did not cause similar malformation. In itself, tamoxifen had no effect on muscle structure or body weight. It should be noted, however, that such effects of tamoxifen were administered to mice with impaired muscle function, that is, to mdx mice (mouse model of Duchenne muscular dystrophy) (*Dorchies et al., 2013*), to FKRPP448L mutant mice with dystroglycanopathy (*Wu et al., 2018*), and to Mtm1 knockout mice (murine model of myotubular myopathy) (*Maani et al., 2018*). The latter is the only study in which wild-type control mice have also been used.

## Role of septins in determining cell proliferation and cell morphology

Complete knockout of Septin7 was not possible in cultured C2C12 cells; however, reduced Septin7 expression significantly decreased proliferation rate and inhibited myotube formation. This is in line with previous observations that septins are essential regulators of cell division from yeast to *Drosophila* (*Kim et al., 2011*; *McMurray et al., 2011*; *Oegema et al., 2000*; *Joo et al., 2007*; *El Amine et al., 2013*; *Kechad et al., 2012*) and in CHO cells (*Oh and Bi, 2011*). In epithelial cells, septin filaments are involved in chromosome movement and spindle elongation during mitosis (*Spiliotis et al., 2005*; *Zhu*

*et al., 2008*), and even cytokinesis of T cells requires the presence of septins (*Mostowy and Cossart, 2012*; *Mujal et al., 2016*).

Furthermore, reduced Septin7 expression resulted in large and nearly circular C2C12 myoblast cells, with a deformed intracellular filamentous septin structure and severely modified mitochondrial network structure. This correlates with the observation that Septin7 determines cell morphology in different cell types, like dorsal root ganglia neurons in the nervous system (*Boubakar et al., 2017*; *Ageta-Ishihara et al., 2013*), or in mouse primary fibroblasts (*Menon et al., 2014*). Finally, involvement of Septin7 as a target gene of miR-127-3p has been revealed in the regulation of C2C12 proliferation (*Li et al., 2020*). These observations, together with our data, suggest that cytoskeletal septin organization has a pivotal role in determining cell morphology, proliferation, and differentiation of skeletal muscle cells. Furthermore, altered cell morphology and filopodia formation in KD cultures also imply the potential role of Septin7 in myoblast migration. Some publications already discussed that modifying the expression of specific septin isoforms (in some cases Septin7) alters the migration of different eukaryotic cell types, like microvascular endothelial cells (*Liu et al., 2014*), human epithelial cells (*Sun et al., 2019*), neural crest cells (*Boubakar et al., 2017*), and human breast cancer or lung cancer cells (*Zhang et al., 2016*; *Zeng et al., 2019*; *Elkhadragy et al., 2020*). The determination of how Septin7 is involved in the migration of myogenic cells is the challenge of future experiments.

## Role of septins in mitochondrial dynamics

Our results suggest the involvement of septins in mitochondrial dynamics in skeletal muscles of mice since mitochondrial area and number were altered following the conditional knockdown of Septin7 expression. Furthermore, long mitochondrial networks were found within these fibers. However, mitochondrial DNA content was reduced, suggesting impaired mitochondrial function in Cre+ mice in line with Afshan and Czajka (*Malik and Czajka, 2013*).

The cytoskeleton has been shown to alter mitochondrial movement and distribution in highly polarized cells and play a role in mitochondrial dynamics (see the review *Pagliuso et al., 2018*). In particular, in septin-depleted cells elongated mitochondria were developed due to decreased fission rather than to a defective mitochondrial fusion (*Pagliuso et al., 2016*). Depletion of Septin-2 has been shown to influence mitochondrial morphology in HeLa cells (*Pagliuso et al., 2016*), and in *Caenorhabditis elegans* (*Pagliuso et al., 2016*), suggesting that the role of septins in mitochondrial dynamics is evolutionarily conserved. The contribution of mitochondria-associated septin cages is also described; for example, Septin7 co-localization with mitochondria was shown to affect the proliferation of *Shigella flexneri* (*Sirianni et al., 2016*), while disrupted mitochondrial morphology was detected in the ciliate *Tetrahymena thermophila* following modification of septin expression (*Wloga et al., 2008*), suggesting that septins maintain mitochondrial stability in ciliates and free-living protists.

Measurements on cultured C2C12 cells suggested that Septin7 is involved in myogenic cell division and myoblast fusion. This in turn indicated, based on previous experiments on satellite cells (see the review *Wang and Rudnicki, 2011*), that it could play an important role in muscle regeneration, too. Indeed, we found (*Figure 6C and E*) that Septin7 expression is increased following a mild muscle injury induced by the injection of BaCl$_2$. The extent of regeneration following the injection was monitored by detecting the presence of Pax7 in satellite cells, and a clear correlation was seen in the time course of the expression pattern of the two molecules. In line with previous observations (*Seale et al., 2000*; *Sambasivan et al., 2011*), the most intense regeneration, and thus the increase in the expression of both Pax7 and Septin7, was seen a few days after the injury, while after 2 weeks regeneration was essentially complete and the expression of the two molecules returned to baseline. As the appearances of Pax7 as well as that of centrally located nuclei are accepted markers of muscle regeneration (*Lepper et al., 2011*; *Mazzotti and Coletti, 2016*), we concluded that the increased expression of Septin7 could also be an essential component of the regeneration process. Although these findings set the basis for the involvement of septin filaments in muscle repair, further experiments are needed to elucidate their exact role in the process.

# Materials and methods

## Key resources table

| Reagent type (species) or resource | Designation | Source or reference | Identifiers | Additional information |
|---|---|---|---|---|
| Gene (*Homo sapiens*) | Human reference genome NCBI | Genome Reference Consortium | http://www.ncbi.nlm.nih.gov/projects/genome/assembly/human | |
| Gene (*Mus musculus*) | Mouse reference genome NCBI | Genome Reference Consortium | http://www.ncbi.nlm.nih.gov/projects/genome/assembly/mouse | |
| Strain, strain background (*Escherichia coli*) | JM 109 competent *E. coli* cells | Promega | Cat# L2005 | |
| Strain, strain background (*M. musculus*) | B6.Cg-Tg(*ACTA1*-cre)79Jme/J mice (ssC0) | Jackson Laboratory | Cat#006149; RRID:IMSR_JAX:006149 | |
| Strain, strain background (*M. musculus*) | C57BL6 *Septin7^{flox/flox}* (SS00) mice | Prof. Dr. Matthias Gaestel | Institute of Physiological Chemistry Hannover Medical School | |
| Cell line (*M. musculus*) | Mouse immortalized C2C12 | ATCC | Cat# CRL-1772; RRID:CVCL_0188 | |
| Biological sample (*Homo sapiens*) | Human *m. quadriceps femoris* | Kenézy Gyula Teaching Hospital of the University of Debrecen | https://kenezykorhaz.unideb.hu/en | |
| Antibody | Anti-Septin7 (rabbit monoclonal) | IBL | Cat# JP18991; RRID:AB_1630825 | IF (1:200), WB (1:250) |
| Antibody | Anti-α-actinin (mouse monoclonal) | Santa Cruz Biotechnology | Cat# sc-17829; RRID:AB_626633 | WB (1:250) |
| Antibody | Anti-Pax7 (mouse monoclonal) | Sigma | Cat# SAB1404168; RRID:AB_10738723 | IF (1:200), WB (1:200) |
| Antibody | Anti-desmin (rabbit polyclonal) | Sigma | Cat# D8281; RRID:AB_476910 | IF (1:200), WB (1:250) |
| Antibody | Anti-Ryanodin receptor 1 (mouse monoclonal) | Thermo Fisher Scientific | Cat# MA3-925; RRID:AB_2254138 | IF (1:250) |
| Antibody | Anti-α-actinin (skeletal muscle specific) (mouse monoclonal) | Sigma | Cat# A7811; RRID:AB_476766 | IF (1:500) |
| Antibody | Anti-myosin4 (MYH4; MF20) (mouse monoclonal) | Thermo Fisher Scientific | Cat# 14-6503-80 | IF (1 µg/ml) |
| Antibody | Anti-L-type calcium channel (rabbit polyclonal) | Santa Cruz Biotechnology | Cat# sc-16229-R | IF (1:250) |
| Antibody | HRP-conjugated anti-rabbit (goat monoclonal) | Bio-Rad Laboratories | Cat# 170-6515; RRID:AB_11125142 | WB (1:500) |
| Antibody | HRP-conjugated anti-mouse (goat monoclonal) | Bio-Rad Laboratories | Cat# 170-6516; RRID:AB_11125547 | WB (1:500) |
| Antibody | Alexa Fluor 488 conjugated anti-rabbit IgG (goat monoclonal) | Thermo Fisher Scientific | Cat# A32731; RRID:AB_2633280 | IF (1:500) |
| Antibody | Cyanine3 conjugated anti-Mouse IgG (goat monoclonal) | Thermo Fisher Scientific | Cat# A10521; RRID:AB_1500665 | IF (1:500) |
| Recombinant DNA reagent | *Septin7*-specific CRISPR/Cas9 KO plasmid constructs | Santa Cruz Biotechnology | Cat# sc-433427 | |
| Recombinant DNA reagent | *Septin7*-specific HDR plasmid constructs | Santa Cruz Biotechnology | Cat# sc-433427-HDR | |
| Recombinant DNA reagent | *Septin7*-specific shRNA constructs in retroviral pGFP-V-RS vectors | Origene | Cat# TR30007 | |
| Commercial assay or kit | Omniscript RT kit | QIAGEN | Cat# 205113 | |
| Commercial assay or kit | RNase free DNase kit | Thermo Fisher Scientific | Cat# EN0521 | |
| Commercial assay or kit | SYBRGreen mix | PCR Biosystems | Cat# PB20.11-51 | |

| Reagent type (species) or resource | Designation | Source or reference | Identifiers | Additional information |
|---|---|---|---|---|
| Commercial assay or kit | High-specificity TaqMan assays | Thermo Fisher Scientific | Cat# 4331182 | |
| Commercial assay or kit | CyQUANT NF Cell Proliferation Assay Kit | Invitrogen | Cat# C35006 | |
| Commercial assay or kit | SuperSignal West Pico PLUS Chemiluminescent Substrate | Thermo Fisher Scientific | Cat# 34577 | |
| Chemical compound, drug | Tamoxifen containing chow | Envigo | Cat# TD 130857 | |
| Chemical compound, drug | Type I collagenase | Sigma | Cat# SCR103 | |
| Chemical compound, drug | Isoflurane | Forane | Cat# NDC 10019-360 | |
| Chemical compound, drug | Durcupan epoxy resin | Sigma | Cat# 44611 | |
| Chemical compound, drug | EZ-Vision Dye 6X | VWR Life Science | Cat# 97064-190 | |
| Chemical compound, drug | MitoTracker Red CMXRos | Thermo Fisher Scientific | Cat# M7512 | (0.5 µM) |
| Chemical compound, drug | Lipofectamine 2000 | Thermo Fisher Scientific | Cat# 11668019 | |
| Chemical compound, drug | Opti-MEM Reduced Serum medium | Thermo Fisher Scientific | Cat# 31985070 | |
| Software, algorithm | Nucline and InterView FUSION software | Mediso Ltd. | http://ctamed.com/en/portfolio/software-solutions/#fusion | |
| Software, algorithm | Axotape software | Axon Instruments | https://www.bioz.com/result/axotape/product/Molecular%20Devices%20Llc | |
| Software, algorithm | Primer Premier 5.0 software | Premier Biosoft | http://downloads.fyxm.net/Primer-Premier-101178.html | |
| Software, algorithm | Program to design target-specific primers for PCR | Primer-BLAST | https://www.ncbi.nlm.nih.gov/tools/primer-blast/ | |
| Software, algorithm | Statistical program Prism | GraphPad Software | https://www.graphpad.com/scientific-software/prism/ | |
| Software, algorithm | Image processing program ImageJ | Image J1.40g | https://imagej.nih.gov/ij/download.html | |
| Software, algorithm | Origin 8.6 | OriginPro | https://originpro.informer.com/8.6/ | |
| Other | DAB substrate kit | Thermo Fisher Scientific | Cat# 34002 | Histological labeling reagent used for the detection of Pax7 expression on muscle sections following $BaCl_2$-induced injury |
| Other | Protein blocking solution serum-free | Dako | Cat# X0909 | Reagent used for blocking unspecific binding sites of the samples before application of primary antibodies in fluorescent immunolabeling procedure |
| Other | TRITC-phalloidin | Sigma | Cat# P1951 | Fluorescently labeled actin-specific labeling IF (1:1000) |
| Other | FITC-phalloidin | Sigma | Cat# P5282 | Fluorescently labeled actin-specific labeling IF (1:1000) |

| Reagent type (species) or resource | Designation | Source or reference | Identifiers | Additional information |
|---|---|---|---|---|
| Other | Mowiol 4-88 | Sigma | Cat# 81381 | Mounting medium used to cover fluorescently labeled samples until the microscopic analysis |
| Other | FACSAria flow cytometer | BD Biosciences | https://www.bdbiosciences.com/en-eu/instruments/research-instruments/research-cell-sorters/facsaria-iii | Device used for single cell sorting based on the expression of fluorescent proteins (GFP and RFP) |
| Other | nanoScan SPECT/CT | Medisol Ltd | http://scanomedtranslational.com/ | Device used for whole body CT scans of mice |
| Other | Mouse running wheel | Campden Instruments Ltd *Bodnár et al., 2014* | | Device used to measure the voluntary activity of mice |
| Other | Capacitive mechano-electric force transducer | Experimetria *Bodnár et al., 2016* | | Device used to measure in vitro muscle force |
| Other | Leica Ultracut UCT | Leica Microsystems | https://www.leica-microsystems.com | Device used to cut ultrathin sections from muscle samples for EM analysis |
| Other | JEM1010 transmission electron microscope | JEOL | https://www.jeolusa.com/PRODUCTS/Transmission-Electron-Microscopes-TEM | Device used to examine ultrathin muscle sections at high magnification |
| Other | HT Mini homogenizer | OPS Diagnostics | Cat# BM-D1030E | Device used to homogenize muscle samples for protein analysis. |
| Other | Labnet MultiGene 96-well Gradient Thermal Cycler | Labnet International | Cat# TC9610-230 | Device used for RT-PCR reactions |
| Other | LightCycler 480 | Roche | Cat# 05015243001 | Device used for quantitative PCR measurements |
| Other | FlexStation 3 multimode microplate reader | Molecular Devices | Cat# Flex3 | Device used to measure fluorescent signal of CyQUANT cell proliferation assay |
| Other | AiryScan Confocal microscope | Zeiss | https://www.zeiss.com/microscopy/int/dynamic-content/news/2014/news-lsm-880.html | Device used to examine fluorescent immunolabeling of muscle fibers and cultured cells at high spatial resolution |

## Experimental model and subject details

### Human subjects
The human study was approved by the Ethics Committee of the Health Science Council, Budapest, Hungary (7917-1/2013/EKU 113/2013). Samples from *m. quadriceps femoris* of human patients going through amputation were taken and used in this study. Amputation surgery and biopsy preparation have been performed at the Kenézy Gyula Teaching Hospital of the University of Debrecen, Hungary.

### Animal care and experimental design
Animal experiments were in compliance with the guidelines of the European Community (86/609/EEC). The experimental protocol was approved by the institutional Animal Care Committee of the

University of Debrecen (2/2019/DEMAB). The mice were housed in plastic cages with mesh covers and fed with pelleted mouse chow and water ad libitum. Room illumination was an automated cycle of 12 hr light and 12 hr dark, and room temperature was maintained within the range of 22–25°C.

### Tamoxifen diet
Tamoxifen diet (*per os*) was started immediately following separation from the mother at age of 4 weeks. Littermates were fed for 3 months without interruption. Chow (Envigo, TD 130857) contains 500 mg tamoxifen/kg diet, providing 80 mg tamoxifen/kg body weight per day assuming 20–25 g body weight and 3–4 g daily intake (*Koitabashi et al., 2009*; *Andersson et al., 2010*).

### Screening of HSA-MCM transgenic lines
B6.Cg-Tg(*ACTA1*-cre)79Jme/J strain originated from the Jackson Laboratory (Bar Harbor, ME). Mice hemizygous for this HSA-Cre transgene are viable, fertile, normal in size, and do not display any gross physical or behavioral abnormalities. These HSA-Cre transgenic mice have the *Cre* recombinase gene driven by the human alpha-skeletal actin (*ACTA1*) promoter. Cre activity is restricted to adult striated muscle fibers. The HSA-MerCreMer (*HSA-MCM*) construct encodes a Cre recombinase (Cre) and contains a mutant estrogen ligand-binding domain (ERT2) that requires the presence of tamoxifen for activity as well. When bred with mice containing a *loxP*-flanked sequence of interest, Cre-mediated recombination will result in striated muscle-specific deletion of the flanked genome. Genomic DNA was isolated from finger biopsies (*Zangala, 2007*) and screened for the presence of the *HSA-MCM* transgene by PCR using the appropriate primers spanned the C-terminus MerCre junction (*Supplementary file 1a*) and produced a 717 bp product.

### *Septin7* genotyping and detection of exon 4 deletion
C57BL/6J *Septin7flox/flox* (SS00) mice were obtained from Prof. Dr. Matthias Gaestel, Institute of Physiological Chemistry Hannover Medical School. These mice were crossed with B6.Cg-Tg(*ACTA1*-cre)79Jme/J mice (ssC0). The SsC0 offspring were back crossed with SS00 mice. Then SSC0 and SS00 mice were taken in breeding to produce litters for tamoxifen feeding.

To investigate the effectivity of Septin7 modification in the skeletal muscle of Cre+*Septin7flox/flox* mice (Cre+ in the following), genomic DNA was prepared from *m. quadriceps femoris*, *m. biceps femoris*, and *m. pectoralis*. Cre-*Septin7flox/flox* mice did not have the MerCreMer construct; these mice were used as littermates. PCR primers were used to differentiate the flox from wild-type mice, and gene deletion was detected by the application of forward2 primer (*Supplementary file 1a*). Based on the presence of specific 151 and 197 bp bands, the wild-type C57BL/6J (BL6 in the following) and floxed mice could be identified, while the deletion of exon 4 proved to be partially achieved as represented by 256 bp bands. According to the gene structure, exon 4 (107 bp, flox region is about 0.5 kb) can be conditionally removed, which results in a non-sense-mediated decay.

## Cell culture
Mouse-immortalized C2C12 myoblast cell line was cultured in high-glucose Dulbecco's Modified Eagle Medium (DMEM) (Biosera, Nuaille, France) supplemented with 10% fetal bovine serum (Gibco by Life Technologies, Carlsbad, CA, USA), 1% Penicillin-Streptomycin (Gibco), and 1% L-glutamine (Biosera). Medium was changed every other day and cells were subcultured at 80–90% confluence. Myoblast fusion and generation of differentiated myotubes were induced by exchanging the culture media to DMEM containing 2% horse serum (Gibco) approximately at 90% confluence.

## CRISPR-Cas9
CRISPR/Cas9 knockout and HDR plasmid constructs specific to mouse *Septin7*, targeting three different places in the coding sequence (in exon# 3, 4, and 5), were purchased from Santa Cruz Biotechnology. Transfection of the C2C12 cells with KO and HDR plasmids was carried out in serum-free Opti-MEM medium (Invitrogen by Thermo Fisher Scientific, OR) using Lipofectamine 2000 transfection reagent (Invitrogen). 48 hr after transfection, puromycin selection was applied for 5 days, then single cells were sorted according to expression of green fluorescent protein (GFP) and red fluorescent protein (RFP) from the KO and HDR vectors, respectively, using FACSAria flow cytometer (BD Biosciences, San Jose,

CA). Single cells were kept in normal culturing media until cell growth was observed in the appropriate cultures. Proliferation was continuously monitored by a transmission microscope.

## Gene silencing

C2C12 cells were seeded in 6-well culture plates in DMEM containing 10% FBS. At 50–70% confluence, medium was replaced by serum-free OptiMEM and cells were transfected with *Septin7*-specific shRNA constructs in retroviral pGFP-V-RS vectors (Origene, Cambridge, UK) using Lipofectamine 2000 transfection reagent. For controls, a noneffective scrambled shRNA cassette vector was employed (Scr). 3 hr after transfection, the medium was replaced by complete DMEM, cells were allowed to recover and synthesize the coded shRNA for 48 hr. Puromycin (2 µg/ml)-containing medium was applied to select cells containing the specific shRNA sequences until well-defined cell clones were visible within the culture plates. Individual clones were then separated and cultured further to analyze the effective gene silencing by Western blot. Appropriate clones presenting significant Septin7 expression change were tested throughout increasing passage numbers, and continuously detectable lower Septin7 expression was required to use the cell clone for further investigation (S7-KD cells).

## In vivo experiments

### In vivo CT imaging

For the in vivo imaging, mice were anesthetized by 3% isoflurane (Forane) with a dedicated small animal anesthesia device. Whole-body CT scans were acquired with the preclinical nanoScan SPECT/CT (Mediso Ltd, Hungary) scanner using the following acquisition parameters: X-ray tube voltage 60 kVp, current 86 mA; exposure time 170 ms per projection; voxel size: $1 \times 1$ mm. For the CT image reconstruction and analysis, the Nucline and InterView FUSION software (Mediso Ltd., Hungary) was used, respectively.

### Voluntary activity wheel measurement

Mice from the different groups (see above) were housed in a cage with a mouse running wheel (Campden Instruments Ltd., Loughborough, UK). Wheels were interfaced to a computer and revolutions were recorded in 20 min intervals, continuously for 14 days. The daily average and maximal speed, distance, and duration of running were calculated for each individual mouse and then averaged by groups (*Supplementary file 1b*).

### Forepaw grip test

The force of forepaw was measured as described earlier (*Bodnár et al., 2014*). Briefly, when the animals reliably grasped the bar of the grip test meter, they were then gently pulled away from the device. The maximal force before the animal released the bar was digitized at 2 kHz and stored by an online connected computer. The test was repeated 10–15 times to obtain a single data point. Measurements for the trained groups were always carried out before the 14 days running regime. For all other animal groups, the grip test was measured on the day when the animals were sacrificed.

## In vitro experiments

### Measurement of muscle force

Muscle contractions were measured as described in our previous reports (*Bodnár et al., 2016*). In brief, fast and slow twitch muscles, *m. extensor digitorum longus* (EDL) and *m. soleus* (Sol), were removed and placed horizontally in an experimental chamber continuously superfused (10 ml/min) with Krebs' solution (containing in mM: 135 NaCl , 5 KCl , 2.5 CaCl$_2$, 1 MgSO$_4$, 10 HEPES, 10 glucose , 10 NaHCO$_3$; pH 7.2; room temperature), equilibrated with 95% O$_2$ plus 5% CO$_2$. One end of the muscle was attached to a rod while the other to a capacitive mechano-electric force transducer (Experimetria, Budapest, Hungary). Two platinum electrodes placed adjacent to the muscle were used to deliver short, supramaximal pulses of 2 ms in duration to elicit single twitches. Force responses were digitized at 2 kHz using TL-1 DMA interface and stored with Axotape software (Axon Instruments, Foster City, CA). Muscles were then stretched by adjusting the position of the transducer to a length that produced the maximal force response and allowed to equilibrate for 6 min.

Single pulses at 0.5 Hz were used to elicit single twitches. At least 10 twitches were measured under these conditions from every muscle. The individual force transients within such a train varied by

less than 3% in amplitude, thus the mean of the amplitude of all transients was used to characterize the given muscle. To elicit a tetanus, single pulses were applied with a frequency of 200 Hz for 200 ms (EDL) or 100 Hz for 500 ms (Sol). Duration of individual twitches and tetani was determined by calculating the time between the onset of the transient and the relaxation to 10% of maximal force.

## Isolation of single skeletal muscle fibers

Single muscle fibers from FDB were enzymatically dissociated in minimal essential media containing 0.2% type I collagenase (Sigma) at 37°C for 30–40 min depending on the muscle weight (*Szentesi et al., 1997*; *Fodor et al., 2008*). To release single fibers, the FDB muscles were triturated gently in normal Tyrode's solution (1.8 mM $CaCl_2$, 0 mM EGTA). The isolated fibers were then placed in culture dishes and stored at 4°C until use.

## Immunofluorescent staining of isolated single fibers

To perform immunocytochemistry, fibers were fixed immediately with 4% paraformaldehyde (PFA) for 20 min. After the fixation method 0.1 M glycine in phosphate buffer saline (PBS) was used to neutralize excess formaldehyde. Fibers were permeabilized with 0.5% Triton-X (TritonX-100, Sigma) for 10 min and blocked with a serum-free protein blocking solution (Dako, Los Altos, CA) for 30 min. Slides were rinsed three times with PBST solution. Primary antibodies (anti-RyR1, anti-Septin7, and skeletal muscle-specific anti-α-actinin, see 'Key resources table') diluted in blocking solution were added to the fibers and slides were incubated overnight at 4°C in a humidity chamber. Samples were washed three times with PBST and incubated with fluorophore-conjugated secondary antibodies for 1 hr at room temperature. After washing three times, a drop of mounting medium was added to each slide (Mowiol 4-88, Sigma) and coverslips placed on the mounting medium. Images from Alexa Fluor 488, TRITC, and DAPI-labeled samples were acquired with an AiryScan 880 laser scanning confocal microscope (Zeiss, Oberkocken, Germany) equipped with a ×20 air and a ×40 oil objective. Excitation at 488, 543, and 405 nm wavelengths was used to detect fluorescence of the aforementioned secondary antibodies, respectively, while emission collected above 550 nm with a long-pass filter.

## Muscle regeneration

Skeletal muscle injury was accomplished by $BaCl_2$ injection. 20 µl of 1.2% $BaCl_2$ (dissolved in physiological saline) was injected to the left *m. tibialis anterior* muscle (TA) of BL6 mice (right *m. tibialis anterior* muscle was nontreated control). Mice were sacrificed 4 and 14 days later followed by removal of TA. Samples were obtained from both injected and noninjected muscles for Western blot analysis and for cryosection-/paraffin-embedded sections. Histological sections were stained with HE or DAB reaction was used to detect Pax7 expression.

## Sample preparation for electron microscopic studies

Freshly prepared TA was fixed in situ with fixative solution (3% glutaraldehyde in Millonig's buffer). Small bundles of fixed muscle fibers were then postfixed in 1% $OsO_4$ in water. For rapid dehydration of the specimens, graded ethanol followed by propylene-oxide intermediate was used. Samples were then embedded in Durcupan epoxy resin (Sigma). Ultrathin horizontal and transversal sections were cut using a Leica Ultracut UCT (Leica Microsystems, Wien, Austria) ultramicrotome and stained with uranyl acetate and lead citrate. Sections were examined with a JEM1010 transmission electron microscope (JEOL, Tokyo, Japan) equipped with an Olympus camera.

EM images were analyzed with ImageJ software, where area, perimeter, aspect ratio, and form factor of the individual myofibrils and mitochondria were determined in the transversal and longitudinal sections from skeletal muscles of Cre- and Cre+ animals.

## RT-PCR analysis

Cell cultures were dissolved, while human muscle biopsies and mouse skeletal muscle were homogenized in Trizol (Molecular Research Center, Cincinnati, OH) with HT Mini homogenizer (OPS Diagnostics, Lebanon, NJ) and subjected to a general RNA isolation protocol. In detail, after the addition of 20% chloroform, samples were centrifuged at 4°C at 16,000×*g* for 15 min. Upper, aqueous phase of samples was incubated in 500 µl (cultures) or 750 µl (tissues) of RNase-free isopropanol at room temperature for 10 min, then total RNA was harvested in RNase-free water and stored at –80°C.

The assay mixture (20 µl) for reverse transcriptase reaction (Omniscript, QIAGEN, Germantown, MD) contained 1 µg RNA, 0.25 µl RNase inhibitor, 0.25 µl oligo (dT), 2 µl dNTP (200 µM), 1 µl high-affinity RT in 10× RT buffer. Amplifications of specific cDNA sequences were carried out using specific primer pairs that were designed by Primer Premier 5.0 software (Premier Biosoft, Palo Alto, CA) based on mouse and human nucleotide sequences published in GenBank and purchased from Bio Basic (Toronto, Canada). The specificity of custom-designed primer pairs was confirmed in silico by using the Primer-BLAST service of NCBI (http://www.ncbi.nlm.nih.gov/tools/primer-blast/). The sequences of primer pairs, annealing temperatures for each specific primer pair, and expected amplimer size for each polymerase chain reaction are shown in *Supplementary file 1a*. Amplifications were performed in a programmable thermal cycler (Labnet MultiGene 96-well Gradient Thermal Cycler; Labnet International, Edison, NJ) with the following settings: initial denaturation at 94°C for 1 min, followed by 30 cycles (denaturation at 94°C, 30 s; annealing at optimized temperatures for each primer pair for 30 s – see *Supplementary file 1a*; extension at 72°C, for 60 s) and then final elongation at 72°C for 5 min. PCR products were mixed with EZ-Vision Dye 6X loading buffer, and DNA bands were visualized following an electrophoresis in 1.2%–2.5% agarose gels.

## Quantitative PCR analysis

Total RNA samples originating from different skeletal muscles (*m. tibialis anterior*, *m. pectoralis*, and *m. quadriceps*) of mice were subjected to DNase treatment and reverse transcription (Thermo Fisher Scientific, Waltham, MA) according to the manufacturer's instructions. Appropriate cDNA samples were used for quantitative PCR reaction (LightCycler 480, Roche, Basel, Switzerland) using either SYBRGreen mix (PCR Biosystems, Oxford, UK) and specific primer pairs for the detection of mitochondrial (*16S RNA*) and nuclear (*Hexokinase*) RNA, or high-specificity TaqMan assays (Mm00550197_m1) against mouse *Septin7* (Thermo Fisher Scientific). All qPCR reactions were conducted in triplicates. $C_p$ values were determined with the Light Cycler 480 SW 1.5.0 software (Roche). Relative copy numbers were calculated via the $\Delta C_p$ method. The ratios of the values of the examined and normalization genes gave the relative expression levels.

## Western blot analysis

Total cell lysates and skeletal muscle tissues were homogenized in a lysis buffer (20 mM Tris–HCl, 5 mM EGTA, Protease Inhibitor Cocktail [Sigma, St. Louis, USA]) with HT Mini homogenizer (OPS Diagnostics). Fivefold concentrated electrophoresis sample buffer (20 mM Tris–HCl, pH 7.4, 0.01% bromophenol blue dissolved in 10% SDS, 100 mM β-mercaptoethanol) was added to total lysates to adjust equal protein concentration of samples and boiled for 5 min at 90°C. 8 or 10 µg of total protein (for Septin7 and Pax7, respectively) were loaded to each lane, and separated in 7.5% SDS–polyacrylamide gel. Proteins were transferred to nitrocellulose membranes, blocked with 5% non-fat milk dissolved in PBS, then membranes were incubated with the appropriate primary antibodies overnight at 4°C (see 'Key resources table'). After washing for 30 min in PBS supplemented with 1% Tween-20 (PBST), membranes were incubated with HRP-conjugated secondary antibodies (see 'Key resources table'). Membranes were developed and signals were detected using enhanced chemiluminescence (Thermo Fisher Scientific). Optical density of signals was measured with ImageJ software (NIH, Bethesda, MD) and results were normalized to the optical density of control tissues.

## Immunofluorescent staining of cultured cells and fluorescent labeling of mitochondria

Mitochondria of cultured cells were labeled with MitoTracker Red CMXRos (0.5 µM) (Thermo Fisher Scientific), in which living cells were incubated for 30 min at 37°C in $CO_2$-thermostated environment. MitoTracker probes passively diffuse across the plasma membrane and accumulate in active mitochondria. Once their mitochondria are labeled, the cells can be treated with an aldehyde-based fixative to allow further processing of the sample. To perform immunocytochemistry, cell cultures were fixed immediately with 4% PFA for 15 min. After the fixation method, 0.1 M glycine in PBS was used to neutralize excess formaldehyde. Fibers were permeabilized with 0.25% Triton-X (TritonX-100, Sigma) for 10 min and blocked with a serum-free protein blocking solution (Dako) for 30 min. Slides were rinsed three times with PBST solution. Primary Septin7 antibody was diluted in blocking solution, added to the samples, and slides were incubated overnight at 4°C in a humidity chamber. On the

next day, samples were washed three times with PBST and incubated with fluorophore-conjugated secondary antibodies and TRITC-phalloidin or FITC-phalloidin (1:1000) for 1 hr at room temperature. After three times washing, a drop of mounting medium was added to each slide. Confocal images were acquired with an AiryScan laser scanning confocal microscope as described above.

## Determination of cellular proliferation

The degree of cellular growth (reflecting proliferation) was determined with CyQUANT NF Cell Proliferation Assay Kit (Invitrogen). C2C12 cells (2500 cells/well) were cultured in 96-well black plates with clear bottoms (Greiner Bi-One, Mosonmagyaróvár, Hungary) for 48 hr. HBSS buffer was prepared (Component C) with deionized water, then one time diluted (1×) dye binding solution was added to the CyQUANT NF dye reagent (Component A). Growth medium was exchanged to 100 µl of 1× dye binding solution. Microplate was covered and incubated at 37°C for 30 min, and fluorescence was measured at 485 nm excitation and 530 nm emission wavelengths using FlexStation 3 multimode microplate reader (Molecular Devices, San Jose, CA). Relative fluorescence values were expressed as percentage of control regarded as 100%.

## Fusion index

Progress of myotube differentiation was quantified using immunocytochemistry in C2C12 cells. Cells were plated into glass coverslips, and in each day of differentiation process samples were fixed in 4% (PFA) solution and subjected to a desmin-specific immunolabeling and DAPI staining. Confocal images were taken from the appropriate samples. Fusion index was calculated as the ratio of the nuclei number in myocytes with two or more nuclei versus the total number of nuclei within the visual fields.

## Quantification and statistical analysis

### Statistical analysis

Pooled data are expressed as mean ± standard error of the mean (SEM). The differences between control mice and animals on tamoxifen diet were assessed using one-way analysis of variance (ANOVA), and all pairwise Bonferroni's multiple comparison method using the statistical program Prism (GraphPad Software, San Diego, CA). *t*-test was used to test the significance, and a p-value of <0.05 was considered statistically significant.

## Acknowledgements

We are grateful to Matthias Gaestel (Institute of Physiological Chemistry, Hannover Medical School, Hannover, Germany) for providing the loxP mice for the experiments. The authors thank Mrs. R Őri for her excellent technical assistance. The authors wish to thank László Balkai, Ildikó Garai, and Scanomed Ltd. (Scanomed Translational Centre, Debrecen, Hungary) for in vivo imaging. This work was supported by the GINOP-2.3.2-15-2016-00044 and the GINOP-2.3.3-15-2016-00020 projects. Project no. TKP2020-NKA-04 has been implemented with the support provided from the National Research, Development and Innovation Fund of Hungary, financed under the 2020-4.1.1-TKP2020 funding scheme. This work was also supported by grants from the Hungarian National Research, Development and Innovation Office (NKFIH K-137600).

## Additional information

### Funding

| Funder | Grant reference number | Author |
| --- | --- | --- |
| National Research, Development and Innovation Fund of Hungary | TKP2020-NKA-04 | László Csernoch |

| Funder | Grant reference number | Author |
|---|---|---|
| National Research, Development and Innovation Fund of Hungary | 2020-4.1.1-TKP2020 | László Csernoch |
| Hungarian National Research, Development and Innovation Office | NKFIH K-137600 | László Csernoch |
| European Union | GINOP-2.3.2-15-2016-00044 | László Csernoch |
| European Union | GINOP-2.3.3-15-2016-00020 | László Csernoch |

The funders had no role in study design, data collection and interpretation, or the decision to submit the work for publication.

## Author contributions

Mónika Gönczi, Formal analysis, Validation, Investigation, Methodology, Writing – original draft; Zsolt Ráduly, László Szabó, Nóra Dobrosi, Formal analysis, Investigation, Methodology; János Fodor, Validation, Investigation, Methodology, Writing – original draft; Andrea Telek, Formal analysis, Investigation, Writing – original draft; Norbert Balogh, Formal analysis, Investigation; Péter Szentesi, Conceptualization, Software, Formal analysis, Validation, Investigation, Methodology, Writing – original draft; Gréta Kis, Investigation, Methodology; Miklós Antal, Conceptualization, Validation, Visualization; György Trencsenyi, Software, Formal analysis, Validation, Investigation; Beatrix Dienes, Conceptualization, Software, Formal analysis, Supervision, Investigation, Visualization, Methodology, Writing – original draft; László Csernoch, Conceptualization, Resources, Supervision, Funding acquisition, Validation, Methodology, Writing – original draft

## Author ORCIDs

Mónika Gönczi ⓘ http://orcid.org/0000-0002-8421-4369
Péter Szentesi ⓘ http://orcid.org/0000-0003-2621-2282
Miklós Antal ⓘ http://orcid.org/0000-0002-2457-7387
László Csernoch ⓘ http://orcid.org/0000-0002-2446-1456

## Ethics

The human study was approved by the Ethics Committee of the Health Science Council, Budapest, Hungary (7917-1/2013/EKU 113/2013). Samples from m. quadriceps femoris of human patients going through amputation were taken and used in this study. Amputation surgery and biopsy preparation have been performed at the Kenézy Gyula Teaching Hospital of the University of Debrecen, Hungary. Animal experiments were in compliance with the guidelines of the European Community (86/609/EEC). The experimental protocol was approved by the institutional Animal Care Committee of the University of Debrecen (2/2019/DEMAB).

## Decision letter and Author response

Decision letter https://doi.org/10.7554/eLife.75863.sa1
Author response https://doi.org/10.7554/eLife.75863.sa2

# Additional files

## Supplementary files

• Supplementary file 1. Informations related to PCR reactions, voluntary running, and in vitro force measurements. (a) Nucleotide sequences, amplification sites, GenBank accession numbers, amplimer sizes, and PCR reaction conditions for each primer pair are shown. Related to *Figure 1*, *Figure 2*, and *Figure 1—figure supplement 1*. (b) Parameters of voluntary running. Related to *Figure 2* and *Figure 2—figure supplement 2*. 3 months of tamoxifen treatment started 4 weeks after birth (4-month-old). 10-day-long running experiment during tamoxifen treatment. *** shows significant difference from Cre- at $p<0.001$ from t-test. (c) Parameters of in vitro force measurement. Related to *Figure 2* and *Figure 2—figure supplement 2*. 3-month tamoxifen treatment started 4 weeks after birth (4-month-old). *, **, and *** show significant difference from Cre- at $p<0.05$, 0.01, and 0.001 from t-test, respectively. #, ##, and ### show significant difference from BL6 at $p<0.05$, 0.01, and

0.001 from t-test, respectively. Fatigue was calculated as the relative amplitude of the 50th tetanus compared to the amplitude of the first tetanus.

- Transparent reporting form

## Data availability

Original Datasets are Generated and provided for the manuscript 'Septin7 is indispensable for proper skeletal muscle architecture and function' (25-11-2021-RA-eLife-75863R2) as: Septin-7 projects: Szentesi, Peter; Csernoch, Laszlo; Gonczi, Monika; Dienes, Beatrix; Fodor, Janos; Telek, Andrea, 2022, https://doi.org/10.48428/ADATTAR/74GBFE. Here we deposited all individual or averaged data used for the generation of figures and supplementary figures; all original data for in vivo and in vitro muscle force measurements; and original CT images.

The following dataset was generated:

| Author(s) | Year | Dataset title | Dataset URL | Database and Identifier |
| --- | --- | --- | --- | --- |
| Peter S, Laszlo C, Monika G, Beatrix D, Janos F, Andrea T | 2022 | Septin-7 projects | https://adattar. unideb.hu/dataset. xhtml?persistentId= doi:10.48428/ ADATTAR/74GBFE | University of Debrecen, 10.48428/ ADATTAR/74GBFE |

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
