## [Editor Report]

This work combines a novel mouse model of inducible skeletal muscle specific deletion of Septin7 with Septin7 manipulation in vitro to explore the role of Septin7 in striated muscle development and function. The work should be of broad interest to muscle and cytoskeletal biologists as it indicates an essential role of Septin7 in normal muscle development and suggests potential roles in muscle regeneration as well.

---

## [Decision Letter]

**Decision letter after peer review:**

Thank you for submitting your article "Septin-7 is indispensable for proper skeletal muscle architecture and function" for consideration by *eLife*. Your article has been reviewed by 3 peer reviewers, one of whom is a member of our Board of Reviewing Editors, and the evaluation has been overseen by Anna Akhmanova as the Senior Editor. The following individual involved in review of your submission has agreed to reveal their identity: Enrique Jaimovich (Reviewer #2).

Essential revisions:

While all 3 reviewers found the work of significant novelty and potential interest, concerns were noted regarding the limited characterization of the novel mouse model that precludes mechanistic understanding of Septin-7's role in muscle physiology or response to injury.

Additional experiments are needed to establish the mechanistic insight and significance of the work, as detailed in the reviewer comments below and summarized here:

1) To determine whether Septin-7's effect is restricted to developing muscle and/or essential in adult muscle homeostasis, the inducible deletion model should be utilized at discrete developmental and fully mature time points.

2) To conclude that Septin-7 plays an important causative role in muscle regeneration following injury, the injury response should be carefully evaluated with and without Septin-7 deletion.

3) Several areas of interest (for example muscle fiber subcellular morphology/mitochondrial phenotypes) require additional quantification and/or more sophisticated analysis.

*Reviewer #1 (Recommendations for the authors):*

1. Examine the effect of septin-7 deletion in adult mice.

2. Examine the effect of septin-7 deletion in developing mice.

3. Examine the effect of septin-7 deletion on regeneration following muscle injury.

4. Provide a more robust characterization of muscle fiber morphological differences after septin-7 depletion, specifically for key components of the EC coupling machinery.

5. Provide a more robust characterization of the mitochondrial morphology differences after septin-7 depletion, including using orthogonal approaches to validate or extend upon the striking decrease in mitochondrial protein content.

*Reviewer #2 (Recommendations for the authors):*

It would be interesting for the readers some additional discussion comparing the loss of Septin-7 in aging skeletal muscle with the loss of muscle function in aging.

[Editors' note: further revisions were suggested prior to acceptance, as described below.]

Thank you for resubmitting your work entitled "Septin7 is indispensable for proper skeletal muscle architecture and function" for further consideration by *eLife*. Your revised article has been evaluated by Anna Akhmanova (Senior Editor) and a Reviewing Editor.

The manuscript has been improved but there is a remaining issue that needs to be addressed, as outlined below:

Conclusive statements regarding the role of Septin 7 in muscle regeneration should be removed. The data show that the expression level of Septin 7 correlates to the muscle regenerative process following injury in WT mice, but this correlation does not demonstrate "a vital contribution (of Septin 7) to muscle regeneration" as stated in the abstract and would require evidence that depletion of Septin 7 in adult mice alters the regenerative process. The impact on myotube differentiation by Septin 7 KO is also no direct evidence of Septin 7 as a critical contribution to muscle regeneration, although it may suggest it. To avoid potential overstatement, please modify the manuscript throughout to state/discuss that Septin 7 may potentially be involved in muscle regeneration following injury in adult muscle, but further studies are needed to characterize its direct role.

---

## [Author Response]

Essential revisions:While all 3 reviewers found the work of significant novelty and potential interest, concerns were noted regarding the limited characterization of the novel mouse model that precludes mechanistic understanding of Septin-7's role in muscle physiology or response to injury.Additional experiments are needed to establish the mechanistic insight and significance of the work, as detailed in the reviewer comments below and summarized here:1) To determine whether Septin-7's effect is restricted to developing muscle and/or essential in adult muscle homeostasis, the inducible deletion model should be utilized at discrete developmental and fully mature time points.

We have elucidated the possibility of inducing Septin7 expression at a number of different stages of intra- and extra-uterine life. If Tamoxifen treatment was started for Cre- pregnant mice, the offsprings were born still. If Tamoxifen treatment was applied for adult animals (where Septin7 expression is reduced as compared to neonatal/young mice) only minor knock-down (less than 20%) could be achieved. These long trials, on the one hand, led us to use the protocol presented in the MS where an approximately 50% reduction in expression could be demonstrated. These findings, on the other hand, are indicative, together with the experiments on cultured cells, that the role of Septin7 is most prominent during muscle development.

2) To conclude that Septin-7 plays an important causative role in muscle regeneration following injury, the injury response should be carefully evaluated with and without Septin-7 deletion.

We understand the concern raised by the Reviewers that a clear causative role of Septin7 in muscle regeneration would require a detailed description of the process in animals with reduced expression of the protein. This in-depth analysis is, in our opinion, beyond the scope of the current study. A thorough analysis with the expression patterns of the key transcription factors and regulatory proteins merits an independent study. Nevertheless, in our manuscript we give evidence that the expression of Septin7 is upregulated (as normalized to α-actinin) following muscle injury clearly indicating that this response is not simply due to the production of new fibers to replace those that were damaged.

3) Several areas of interest (for example muscle fiber subcellular morphology/mitochondrial phenotypes) require additional quantification and/or more sophisticated analysis.

A more detailed analysis of subcellular morphology including that of mitochondria has been carried out. This includes immunostaining for actin, myosin, and the L-type calcium channel, staining of the T-tubules on freshly isolated muscle fibers. The formers are now presented in Figure 2—figure supplement 3. Mitochondrial morphology was assessed by calculating key parameters which led to the reorganization of the corresponding figure (Figure. 5) in the revised manuscript.

Reviewer #1 (Recommendations for the authors):1. Examine the effect of septin-7 deletion in adult mice.2. Examine the effect of septin-7 deletion in developing mice.

Since there was no previous information about the expression of septin isoforms in skeletal muscles and about its changes during development, in the beginning of our experiments we collected data in this regard. Septin7 protein expression was determined in skeletal muscle samples from mice at different developmental stages. As presented in Figure. 1C we observed a decrease in Septin7 protein expression from newborn to adult stages. The expression profile of Septin7 was also investigated in samples from 2, 4, 6, 9, and 18-month-old mice. Here, we observed a significant decrease in Septin7 protein expression in samples isolated from mice of 4, 6, 9, and 18 months of age (58±8; 48±9; 66±16; 54±9% relative to the 2-month-old muscles, respectively), however there were no considerable changes between samples after 4 months of age.

In order to generate skeletal muscle specific, conditional Septin7 knockdown animals, we applied Tamoxifen treatment at different developmental stages in our preliminary studies (see Author response table 1; Author response image 1; Author response image 2) . When Cre- pregnant females were fed with Tamoxifen in the third trimester of pregnancy, it caused intrauterin lethality independent of the genotype. According to the animal ethics requirements we did not continue this experimental protocol.

**Author response image 1. sa2fig1:** Figure represents the presence of floxed sites at Septin7 gene (white arrow) and the deletion of exon 4 (red arrows) in the appropriate DNA samples isolated from mice treated with Tamoxifen from different age and using different methods and periods of Tamoxifen application. Exon 4 deletions were less than 20%, therefore these trials were not continued. Numbers above each lane correspond to the animal ID-s presented in the table above. Q – m. quadriceps, B- m. biceps femoris, P – m. pectoralis.

**Author response image 2. sa2fig2:** Images representing the T-tubule system of a single muscle fibers isolated from Cre- and Cre+ FDB muscle. Di-8-ANEPPS staining reveal no considerable difference between the two type of animals suggesting that the reduction of Septin7 expression do not alter the T-tubular system of skeletal muscle cells.

**Author response table 1. sa2table1:** Genetic modification of Septin7 gene following Tamoxifen treatment in mice mentioned below.

mouse ID(genotype)	age at beginning of treatment	duration of treatment	type of treatment	sacrifice
**43** **(Cre+)**	3-month-old	5 consecutive days 24h interval	20mg/ml Tamoxifen injection (100µl)	10 days after injection
**59** **(Cre+)**	3-month-old	5 consecutive days 24h interval	20mg/ml Tamoxifen injection (100µl)	17 days after injection
**73** **(Cre+)**	3-month-old	5 consecutive days 24h interval	40mg/ml Tamoxifen injection (100µl)	10 days after injection
**79** **(Cre+)**	3-month-old	2X5 consecutive days 24h interval	20mg/ml Tamoxifen injection (100µl)	10 days after injection
**108** **(Cre+)**	3-month-old	4 weeks	80mg/kg body weight Tamoxifen diet	after diet termination
**116** **Cre+**	3-month-old	4 weeks	80mg/kg body weight Tamoxifen diet	5 weeks after diet termination
**120** **Cre+**	3-month-old	4 weeks	80mg/kg body weight Tamoxifen diet	6 weeks after diet termination
**10** **Cre+**	1-month-old	6 weeks	80mg/kg body weight Tamoxifen diet	after diet termination
**16** **Cre+**	1-month-old	6 weeks	80mg/kg body weight Tamoxifen diet	after diet termination
**18** **Cre+**	1-month-old	6 weeks	80mg/kg body weight Tamoxifen diet	after diet termination
**19** **Cre-**	1-month-old	6 weeks	80mg/kg body weight Tamoxifen diet	after diet termination

In the next stage of our initial experiments, 3-month-old mice were treated with both intraperitoneal injections for 5 consecutive days or Tamoxifen diet for 4 weeks. Here, only a moderate deletion of the exon4 was detected in *Septin7* gene in Cre+ animals (data obtained from these mice are shown in Author response table 1). These findings and the observation of ontogenesis dependent expression of Septin7 indicated its significance at the early stage of development and suggested that we should modify the gene expression at earlier age. Six weeks of diet supplemented with Tamoxifen generated well detectable exon deletion in younger (1-month-old) mice. Based on these observations we decided to start the Tamoxifen-supplemented diet in younger (4-week-old) animals immediately after separation from the mother and we kept the treatment for a longer period (3 months) to be sure that exon deletion will be prominent in all Cre+ animals.

3. Examine the effect of septin-7 deletion on regeneration following muscle injury.

We accept the concern of the Reviewer that the effect of Septin7 deletion on muscle regeneration should be investigated in more detail and that those results would further prove the role of this protein in skeletal muscle. We, therefore, already started to investigate the regeneration process in Septin7 depleted skeletal muscle. BaCl_2_ injection was used to initiate a mild injury and the accompanying changes in morphology and the expression of different regulatory proteins were examined. We, however, believe that these experiments not only require more time despite the advances already made but are of substantial size that these data should not be included into this manuscript (which is already at its limit for space), rather they should be presented as an independent manuscript.

4. Provide a more robust characterization of muscle fiber morphological differences after septin-7 depletion, specifically for key components of the EC coupling machinery.

We are grateful for the Reviewer's suggestion about further characterization of muscle fiber morphological changes in Septin7 depleted samples, however, one needs to be aware that the extent of the decrease in Septin7 expression in the individual skeletal muscle fibers would not necessarily be equal.

Nevertheless, additional immunolabeling on isolated single fibers from FDB muscles of Cre+ and Cre- mice were conducted. We now present confocal images showing the results from actin and MYH4 (as two main contractile proteins), and L-type calcium channel-specific staining from Cre+ and Cre- samples (see the Figure 2—figure supplement 3 in the revised version) as well. Double immunolabeling was carried out alongside Septin7. We conclude from these experiments (and from the EM images presented in Figures. 4 and 5), that significant difference in the key components of skeletal muscle contractile elements and one of the main EC coupling channel proteins were not observed between the muscle fibers of the two groups.

Furthermore, intact FDB fibers were incubated with Di-8-ANEPPS dye to label the T-tubule system. As the representative images below demonstrate, we could not detect any significant alteration of T-tubules in Septin7 modified Cre+ animal samples as compared with its control Cre- counterpart. Unfortunately, for technical reasons the double labeling of T-tubule system with Di8-ANEPPS and Septin7 immunostaining was not successful.

5. Provide a more robust characterization of the mitochondrial morphology differences after septin-7 depletion, including using orthogonal approaches to validate or extend upon the striking decrease in mitochondrial protein content.

Indeed, there was a discrepancy between the representative images (Figure 5E and 5F) indicating differences in mitochondrial morphology between Cre+ and Cre- mice and the data shown in Figure 5C. We thank the Reviewer for calling our attention to this point. We reexamined the data based on the suggestions made by the Reviewer (see below) and reorganized the panels in Figure 5. Please note that the average data in the original MS were determined in transversal EM sections, while panel F of the original figure (now panel G) presents the large mitochondrial network in horizontal sections of Cre+ muscles. Also, in Figure 5G of the original manuscript (now panel I), individual measured data of total area of mitochondria within the examined visual fields represent two, well separated data sets, one of which has similar value as compared with the Cre-, while the other set represents significantly higher total area causing significantly increased median of the averaged data.

Nevertheless, as suggested, a more detailed morphological characterization of the mitochondria was carried out and thus the aspect ratio (AR) and form factor (FF) values of the mitochondria were also determined using the same transversal EM images. In Cre+ samples both parameters (AR=1.82±0.04; FF=1.53±0.03) were altered as compared with the Cre- data (AR=2.04±0.03; FF=1.49±0.02), but the difference was statistically significant only in the case of AR data (p<0.001 and p=0.16, respectively). Both of the aforementioned parameters were significantly higher in Cre+ samples in the longitudinal sections.

Furthermore, changes in mitochondrial morphology following Septin7 depletion have been further investigated in cultured C2C12 cells using MitoTracker Red CMXRos staining in living cells (see Figure 5—figure supplement 2 in the revised version).

Reviewer #2 (Recommendations for the authors):It would be interesting for the readers some additional discussion comparing the loss of Septin-7 in aging skeletal muscle with the loss of muscle function in aging.

We are grateful for the reviewer's appreciation regarding our results and its importance in muscle physiology. We accept the concern of the Reviewer that the comparison of Septin7 deletion and the loss of muscle function with aging would be essential in the discussion. Accordingly, we extended the Discussion of the revised manuscript to address this.

We, however, believe that our results indicate more profound role of Septin7 in the earlier developmental phase of skeletal muscle. Nonetheless, our preliminary results suggest that cytoskeletal septins (here, Septin7 was investigated) could play an important role in muscle regeneration and in the formation of new myofibers. Further investigations are required to describe the exact mechanism behind this. In addition, we have already started experiments regarding the functional role of Septin7 in the migration of myogenic cells. These experiments are in our view, beyond the scope of the current manuscript but could be the basis of further studies.

[Editors' note: further revisions were suggested prior to acceptance, as described below.]

The manuscript has been improved but there is a remaining issue that needs to be addressed, as outlined below:Conclusive statements regarding the role of Septin 7 in muscle regeneration should be removed. The data show that the expression level of Septin 7 correlates to the muscle regenerative process following injury in WT mice, but this correlation does not demonstrate "a vital contribution (of Septin 7) to muscle regeneration" as stated in the abstract and would require evidence that depletion of Septin 7 in adult mice alters the regenerative process. The impact on myotube differentiation by Septin 7 KO is also no direct evidence of Septin 7 as a critical contribution to muscle regeneration, although it may suggest it. To avoid potential overstatement, please modify the manuscript throughout to state/discuss that Septin 7 may potentially be involved in muscle regeneration following injury in adult muscle, but further studies are needed to characterize its direct role.

We have revised the manuscript according to the Senior Editor´s and Reviewing Editor´s suggestion and statements regarding the role of Septin7 in muscle regeneration have been reworded.

The following changes were made in the main text:

Abstract; page 2, line 36-37

Introduction; page 4, line 92-93

Results; page 12, line 297, 311

Discussion; page 17, line 437.

Hope, that the manuscript with these aforementioned changes, *i.e.* only suggesting a role for the cytoskeletal protein Septin7 in muscle regeneration is now in an acceptable form.